# High-throughput quantum photonic devices emitting indistinguishable photons in the telecom C-band

Paweł Holewa [1,2,3] ✉, Daniel A. Vajner [4], Emilia Zięba-Ostój [1], Maja Wasiluk [1], Benedek Gaál [2], Aurimas Sakanas[2], Marek G. Mikulicz [1], Paweł Mrowiński[1], Bartosz Krajnik [1], Meng Xiong [2,3], Kresten Yvind [2,3], Niels Gregersen [2], Anna Musiał [1], Alexander Huck [5], Tobias Heindel [4], Marcin Syperek [1,6] ✉ & Elizaveta Semenova [2,3,6] ✉

Single indistinguishable photons at telecom C-band wavelengths are essential for quantum networks and the future quantum internet. However, high-throughput technology for single-photon generation at 1550 nm remained a missing building block to overcome present limitations in quantum communication and information technologies. Here, we demonstrate the high-throughput fabrication of quantum-photonic integrated devices operating at C-band wavelengths based on epitaxial semiconductor quantum dots. Our technique enables the deterministic integration of single pre-selected quantum emitters into microcavities based on circular Bragg gratings. Respective devices feature the triggered generation of single photons with ultra-high purity and record-high photon indistinguishability. Further improvements in yield and coherence properties will pave the way for implementing single-photon non-linear devices and advanced quantum networks at telecom wavelengths.

A quantum network[1] based on remote nodes interconnected via fiber-optical links and capable of transferring quantum information using flying qubits will provide the backbone for the implementation of protocols for secure communication[2,3] and distributed quantum computing[4]. Notably, the network can rely on the existing silica-fiber-based infrastructure, utilizing a low-loss channel for the transmission of photons with a wavelength in the telecom C-band around 1550 nm[5]. These quantum network architectures can benefit from existing components and classical signal management protocols, hence making it feasible to transfer quantum information over large distances[5].

In recent years, the technology for the epitaxial growth of self-assembled quantum dots (QDs) has rapidly advanced, resulting in the demonstration of QD-based single-photon sources (SPSs) with excellent characteristics. These include high photon extraction efficiencies (~79%)[6], high single photon generation rates ( ~ 1GHz)[7], and near unity photon indistinguishability ( > 96%)[7,8], however, all achieved outside the telecom-relevant C-band. Besides the extraordinary material quality, these characteristics are achieved owing to efficient light-matter coupling between a QD and a suitable photonic element. For efficient coupling, spectral and spatial matching is required between the quantum emitter and the engineered photonic mode, which is

[1]Department of Experimental Physics, Faculty of Fundamental Problems of Technology, Wrocław University of Science and Technology, Wyb. Wyspiańskiego 27, 50-370 Wrocław, Poland. [2]DTU Electro, Department of Electrical and Photonics Engineering, Technical University of Denmark, Ørsteds Plads 343, DK-2800 Kongens Lyngby, Denmark. [3]NanoPhoton - Center for Nanophotonics, Technical University of Denmark, Ørsteds Plads 345A, DK-2800 Kongens Lyngby, Denmark. [4]Institute of Solid State Physics, Technische Universität Berlin, 10623 Berlin, Germany. [5]Center for Macroscopic Quantum States (bigQ), Department of Physics, Technical University of Denmark, DK-2800 Kongens Lyngby, Denmark. [6]These authors contributed equally: Marcin Syperek, Elizaveta Semenova. ✉e-mail: pawel.holewa@pwr.edu.pl; marcin.syperek@pwr.edu.pl; esem@fotonik.dtu.dk

challenging due to the spatial and spectral distribution of epitaxial QDs. Until now, the QD coupling to photonic cavities operating around 1550 nm has only been realized using non-deterministic fabrication processes, limiting device yield and scalability[9].

In this article, we report on the high-throughput fabrication of nanophotonic elements around pre-selected individual QDs emitting single and indistinguishable photons in the telecom C-band. For this purpose, we develop a near-infrared (NIR) imaging technique for self-assembled InAs/InP QDs utilizing a hybrid sample geometry with enhanced out-of-plane emission from single QDs[10] and a thermo-electrically cooled InGaAs camera in a wide-field imaging configuration. In combination with two electron-beam lithography (EBL) steps, our method enables an overall positioning accuracy of 90 nm of the QD with respect to the photonic element and allows for rapid data collection as compared to competing techniques based on scanning in-situ imaging[11]. We apply our technique for the deterministic integration of pre-selected QDs into circular Bragg grating (CBG) cavities. The proposed technological workflow allows us to greatly enhance the device fabrication yield reaching ~30%, which is a significant improvement compared to <1% that would typically be achieved with a random placement approach. The QD-CBG coupling is evidenced by a Purcell factor ~5, and our devices demonstrate a state-of-the-art photon extraction efficiency of $\eta = (16.6 \pm 2.7)\%$ into the first lens with a numerical aperture (NA) of 0.4, a high single-photon purity associated with $g^{(2)}(0) = (3.2 \pm 0.6) \times 10^{-3}$, and a record-high photon-indistinguishability of $V = (19.3 \pm 2.6)\%$ for QD-based SPSs at C-band wavelengths.

## Results

### Design of a QD structure for wide-field imaging

Imaging at very low light levels at wavelengths > 1 µm is challenging due to the high level of electronic noise of respective camera systems. Although cameras based on InGaAs achieve quantum efficiencies > 80%, they are characterized by a factor of >$10^4$ higher dark currents compared to Si-based devices. The photon emission rate from the sample is therefore of key importance for the ability to image and localize single QDs. Following our previous work[10], we have designed a planar sample geometry that significantly enhances the photon extraction efficiency, allowing to localize single QDs and the subsequent fabrication of photonic elements.

The planar QD structure consists of an epitaxially grown 312 nm-thick InP slab containing a single layer of InAs QDs. The InP slab is atop a 359 nm-thick $SiO_2$ layer with a bottom Al mirror bonded to a Si wafer carrier (Fig. 1a, see Methods). Overall, this geometry enhances the QD emission in the out-of-plane direction by a factor >7 as compared to bulk InP samples, reaching a total photon extraction efficiency of >10% from a single QD for NA = 0.4[10]. In fact, this design turned out to be crucial for the imaging step, as the SNR of the QD emission was

insufficient for investigated structures without a backside mirror (see Supplementary Note 3). For QD localization later in the experiment, we structure the top InP layer in a mesh with fields of size $(50 \times 50)$ µm$^2$ separated by 10 µm (see Methods), where the field edges are used as alignment marks (AMs) for imaging. The fields are furthermore organized in blocks accompanied by InP crosses that allow us to align the electron beam to specific target QDs during the EBL process (see Supplementary Note 2 for the optical microscope image of the sample surface with fabricated cavities).

For the self-assembled Stranski-Krastanov QD epitaxy, we employed the near-critical growth regime in metalorganic vapor-phase epitaxy (MOVPE)[12] (see Methods), and obtained a QD surface density of $3.1 \times 10^8$/cm$^2$ corresponding to an average QD separation of 1.5 µm. Since QDs exhibit a size, shape, and strain distribution, only a fraction of the QDs have their ground-state optical transition in the C-band. With a $(1550 \pm 8)$ nm bandpass filter, we find on average $N_F = 10$ QDs per field, which translates to an effective QD density of $4 \times 10^5$/cm$^2$ and an average QD separation of ~16 µm.

### Design of circular Bragg grating cavities

The CBG geometry is optimized using the modal method (see Supplementary Note 1) to enhance the cavity figures of merit at 1550 nm, namely the collection efficiency at the first lens and the Purcell factor ($F_P$). As opposed to other implementations[9], we consider a simplified CBG geometry consisting of a central mesa and only four external rings. According to our calculations, this number is sufficient for high $\eta$ and $F_P$, providing a smaller footprint and less complexity in the fabrication process. The in-plane cavity dimensions include the central mesa radius of $R_0 = 648$ nm, the grating period of 747 nm, and the separation between InP rings (air gap) of 346 nm. For these geometrical parameters, Fig. 1b shows the calculated broadband $\eta$ that amounts to nearly 62% and 82% at 1550 nm for NA of 0.4 and 0.65, respectively, which is similar to other state-of-the-art CBG designs[13–15]. The wavelength dependence of the Purcell factor, presented in Fig. 1b, mimics the CBG cavity mode centered at 1550 nm and reaches a maximum value $F_P = 18.1$ with a quality factor of 110. The influence of the cavity geometry on the dispersion of $\eta$ and $F_P$ suggests that the cavity grating together with the Al mirror creates a photonic bandgap that governs the $\eta$-dependence and enhances the emission directionality, while the InP membrane thickness and central mesa diameter crucially affect the center wavelength of the $F_P$-dependence. A scanning electron microscopy (SEM) image of a fabricated CBG cavity is shown in Fig. 1c.

### Optical imaging and QD localization

The NIR imaging setup utilizes a wide-field bright microscope configuration as shown in Fig. 2a. The structure with QDs above the Al

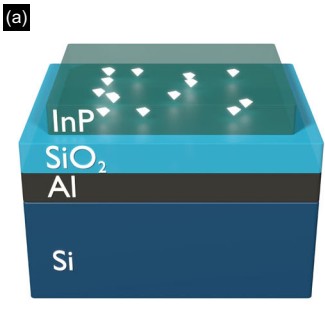
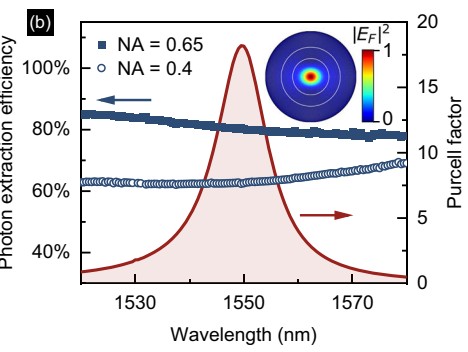
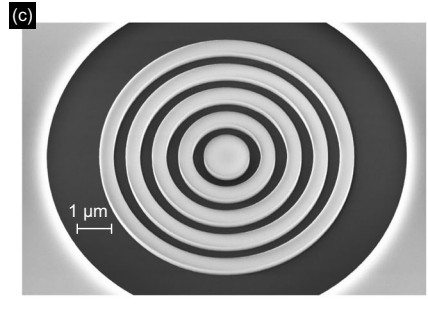

**Fig. 1 | Structure with quantum dots (QDs) for imaging and optimized circular Bragg gratings (CBGs). a** The layer stack for efficient localization of single InAs/InP QDs emitting at C-band. **b** Calculated CBG Purcell factor (dark red line, right axis) and photon extraction efficiency (numerical aperture (NA) = 0.65, full squares, and NA = 0.4, empty circles, left axis). Inset: far-field emission pattern with rings marking NA = 0.4 and NA = 0.65. $E_F$ - normalized electric field amplitude. **c** Scanning electron microscope image of a CBG cavity etched in InP on top of $SiO_2$.

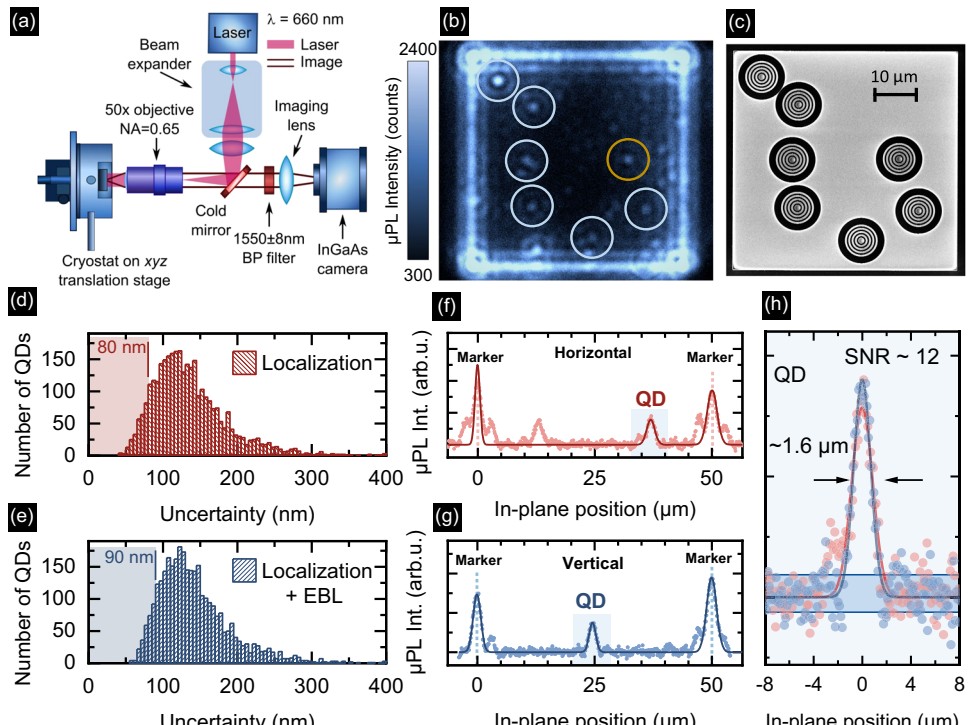

**Fig. 2 | Microphotoluminescence imaging of quantum dots (QDs). a** The optical setup used for imaging, BP - bandpass filter, NA - numerical aperture, **b** a microphotoluminescence map of a $(50 \times 50)\,\mu m^2$ InP field containing seven localized InAs QDs emitting at C-band, **c** scanning electron microscope image of CBGs fabricated atop the localized and preselected QDs, **d**, **e** histograms of QD localization accuracy (**d**) and overall cavity placement accuracy (**e**) for all detected spots, with markers for $10^{th}$ distribution percentiles, **f**, **g** exemplary microphotoluminescence map cross-sections showing the signal of the QD labeled with the orange circle in (**b**) and alignment marks together with Gaussian fits used for the QD localization, **h** close-up of the QD signal from (**f**) and (**g**) centered to zero. The full width at half maximum of the fitted Gaussian profiles is ~1.6 μm and the signal-to-noise ratio (SNR) is 12.

reflector is mounted in an optical cryostat at $T = 4.2$ K movable by an x-y-z stage for targeting fabricated fields that are imaged consecutively. For sample illumination, we use a 660 nm continuous-wave (CW) semiconductor laser diode, spatially shaped with a beam expander, and focused on the backside of a commercially-available microscope objective (NA = 0.65) with 50 × magnification, 57% transmission in the NIR, and 10 mm working distance. This configuration provides nearly homogeneous surface illumination across a $(50 \times 50)\,\mu m^2$ field and high photon collection efficiency. The spatially-distributed QD microphotoluminescence (μPL) and scattered light from the field edges (here used as AMs) are collected by the same objective and pass through a cold mirror cutting off the laser light. Finally, the emission is projected onto a thermo-electrically cooled InGaAs-based camera with a $(12.8 \times 10.24)\,mm^2$ chip and a pixel size of $(20 \times 20)\,\mu m^2$. With the 4 × magnification lens in front of the camera, the setup has a 200 × magnification, enabling the optimal filling of the entire camera chip with a single field (Fig. 2b). The 1550 nm band-pass filter with 8 nm full-width half-maximum (FWHM) placed in front of the imaging lens selects QDs with emission in the C-band.

Figure 2b shows a representative image of a field recorded with a camera integration time of 2.5 s. The QDs can clearly be recognized as individual bright spots with FWHM ≈ 1.6 μm (see Fig. 2h and Supplementary Note 3) and Airy rings around. The square-shaped outline of the field scatters light and is used as AM for QD localization.

The localization of QDs is performed by taking vertical and horizontal cross-sections both crossing at a QD emission spot in the μPL intensity map. Each cross-section thus contains the position of the target QD relative to two AMs (Fig. 2f, g). Gaussian profiles fitted to the QD and the AM signals are subsequently used to determine the QD peak position relative to the AMs. The average signal-to-noise ratio for QD emission spots is 10.6, emphasizing the importance of the

7 × emission enhancement in the planar structure as compared to bulk InP. We find that for the brightest 10% of all QDs with SNR > 15.5, the position is fitted with an uncertainty of <54 nm in 1D and with an uncertainty of the AM position of <36 nm, resulting in a total uncertainty of QD position in 1D of <62 nm. In 2D, this translates to 80 nm accuracy for the QD localization. Finally, taking into account the EBL alignment accuracy of 40 nm as measured in our previous work[16], we estimate the overall accuracy of 2D CBG placement to $\Delta R = 90$ nm. The histograms of QD localization accuracy and overall cavity placement accuracy for all detected spots are shown in Fig. 2d and Fig. 2e respectively, and the 80 nm and 90 nm levels are marked for reference. Medians for the distributions are slightly larger, 127 nm and 133 nm, due to processing of all detected spots, irrespective of their brightness and expected cavity fabrication precision. Details on the localization algorithm, derivation and discussion of the uncertainties, and data on the accuracy of cavity positioning are given in the Supplementary Note 3. Following the localization of suitable QDs, CBGs are fabricated using EBL with proximity error correction and an optimized inductively coupled plasma-reactive ion etching (ICP-RIE) process (see Methods). The SEM image presented in Fig. 2c is taken from the same field after fabricating the CBG cavities around pre-selected QDs indicated by the circles in Fig. 2b.

## Deterministic process yield

We use a μPL setup to quantify the process yield that we define as the ratio between the number of QD-CBG devices with QD emission spectra matching the CBG mode and the number of all CBGs investigated, and we obtain $Y = 30\%$. This value should be compared with the yield that would be obtained in a statistical QD-CBG fabrication process. As we estimate the average number of QDs per field of size $F = 50\,\mu m$ to $N_F = 10$, the probability of covering one of them with the

central mesa of diameter $2R_0$ is $Y_{random} \sim N_F \times (2R_0/F)^2 = 0.67\% \ll Y$. Some of the QDs (inside a CBG) emission spectra are significantly broadened (median linewidth of 0.76 nm, see Supplementary Note 6) as compared to the narrowest recorded linewidth of 0.14 nm (identical to the spectrometer resolution). We attribute the broadening to the impact of surface states and point defects caused by the cavity fabrication, effectively resulting in the spectral wandering of the QD emission line[17]. Such defects can also introduce non-radiative recombination centers in the close vicinity of or even into the QD, quench the optical emission, and effectively reduce the process yield. Using the temperature-dependent $\mu$PL studies, we make sure that even the broadened emission lines follow the expected Varshni trend, ensuring that these spectral lines can indeed be associated with the QD emission as the temperature dependence of the cavity mode energy is much weaker (see Supplementary Note 6).

## Purcell enhancement

In the following, we discuss the optical properties of two exemplary devices, QD-CBG #1 and QD-CBG #2, each containing a single pre-selected QD coupled to the CBG cavity mode (see Supplementary Note 6 for the properties of a third device QD-CBG #3). Figure 3a shows the narrow QD emission lines overlaid on the cavity mode with $Q = 194$, the latter obtained under high power cavity excitation, evidencing good spectral overlap between the cavity mode and the QD emission. We interpret the dominant QD emission lines in both devices as trions (CX), due to their linear intensity dependence on excitation power, and the lack of fine-structure splitting. This is in line with typical spectra for our InAs/InP QDs with preferential CX recombination where the average CX binding energy was measured to be 4.7 meV[10].

The coupling between the QD and the CBG cavity is evidenced by the observation of a reduced emission decay time as compared to the decay of QDs in the planar reference structure. For the CX line in QD-CBG #1 and #2 we record decay times of $\tau_{\#1} = (0.40 \pm 0.01)$ ns and $\tau_{\#2} = (0.53 \pm 0.01)$ ns, respectively (Fig. 3b). To take statistical QD-to-QD fluctuations for the reference decay time into account, we estimate the average decay time of 8 QD CX lines observed from dots located outside of the cavities, i.e. without Purcell-induced modification of the radiative lifetime, and obtain $\tau_{ref} = (1.99 \pm 0.16)$ ns (see Supplementary Note 6), while the single reference shown in Fig. 3b has a decay time of $(2.01 \pm 0.02)$ ns. Therefore, the measured Purcell factor for QD-CBG #1 is $F_P = (5.0 \pm 0.4)$ and $F_P = (3.8 \pm 0.3)$ for QD-CBG #2. Although the obtained Purcell factors are comparable with $F_P = 3$ obtained in the non-deterministic fabrication approach[9], we expect it to be much higher if the QD perfectly matched the cavity mode both spectrally and spatially. However, the expected Purcell factor decays rapidly with the

dipole displacement $r_0$ from the cavity center (decreases by half for $|r_0| = 100$ nm, see Supplementary Note 1), as the fabricated CBG is optimized for a higher-order mode that exhibits electric field minima along the cavity radial direction (see Supplementary Note 1). Hence, the relatively large total positioning uncertainty for QD-CBGs #1–#3 ($\Delta R \sim 140$–150 nm) and the non-ideal spectral emitter-cavity overlap explain the reduced $F_P$ as compared to the model.

## Photon extraction efficiency

We evaluate the photon extraction efficiency by recording the power-dependent $\mu$PL signal with a superconducting nanowire single-photon detector (SNSPD) in a calibrated optical setup (see Fig. 3c). The setup has a total transmission of $(1.1 \pm 0.2)\%$ (see Supplementary Note 5). The measured $\eta$ values are corrected by the factor $\sqrt{1 - g^{(2)}(0)}$, to account for the detection of secondary photons due to the refilling of QD states[18,19]. Here, the $g^{(2)}(0)$ value is obtained under the excitation power $P_{sat}$ corresponding to saturation of the CX line. Evaluating the CX emission, we obtain an extraction efficiency $\eta_{\#1} = (16.6 \pm 2.7)\%$ for QD-CBG #1 and $\eta_{\#2} = (13.3 \pm 2.2)\%$ for QD-CBG #2 using an objective with NA = 0.4.

## Single-photon emission purity

The photon statistics of a quantum light source is of fundamental importance for applications in photonic quantum technologies. In the following, we investigate the single-photon purity of the emission from QD-CBG #2 by analyzing the photon autocorrelation function $g^{(2)}(\tau)$ (cf. Supplementary Note 7 for details on the data analysis and complementary $g^{(2)}(\tau)$ measurements).

Figure 4 depicts the measured $g^{(2)}(\tau)$ histograms obtained under pulsed off-resonant excitation at a power $0.5 \times P_{sat}$ (Fig. 4a) and LO-phonon-assisted, quasi-resonant excitation at $0.04 \times P_{sat}$ (Fig. 4b). Under off-resonant excitation, the single-photon purity is limited by recapture processes resulting in $g^{(2)}(0)_{fit} = (0.05 \pm 0.02)$, where the uncertainty is mainly determined by the background level $B$. From the fit, we determine a decay time of $\tau_{dec} = (0.67 \pm 0.03)$ ns, in good agreement with the spontaneous emission decay time observed in Fig. 3b, $\tau_{\#2} = (0.53 \pm 0.01)$ ns.

Under weak quasi-resonant excitation at $P \ll P_{sat}$, the probability for charge-carrier recapture is strongly reduced, resulting in almost negligible background contributions ($B = 0$) (Fig. 4b) and a fitted value of $g^{(2)}(0)_{fit} = (4.7 \pm 2.6) \times 10^{-3}$ at $P = 0.04 \times P_{sat}$. Additionally, we evaluated the raw antibunching value by integrating the raw coincidences around $\tau = 0$ over a full repetition period normalized by the Poisson level of the side peaks. This results in $g^{(2)}(0)_{raw} = (3.2 \pm 0.6) \times 10^{-3}$, with the error deduced from the standard deviation of the distribution of counts in the side peaks. As discussed later, this result compares

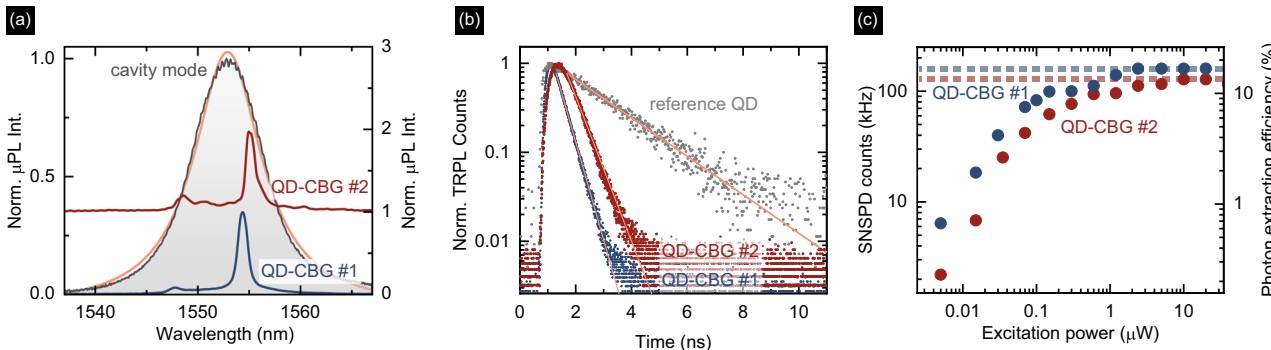

**Fig. 3 | Characteristics of exemplary fabricated quantum dot-circular Bragg grating (QD-CBGs) devices #1 and #2. a** Microphotoluminescence ($\mu$PL) spectra for QDs in devices #1 and #2 overlaid on the cavity mode of device #2 (gray) and fitted with a Lorentzian profile (orange), stacked for clarity. **b** Time-resolved $\mu$PL (TRPL) traces for these QDs with the reference QD decay. **c** Power-dependent count rates registered on the superconducting nanowire single-photon detector (SNSPD). Horizontal lines mark the line $\mu$PL signal saturation level used for the determination of photon extraction efficiency.

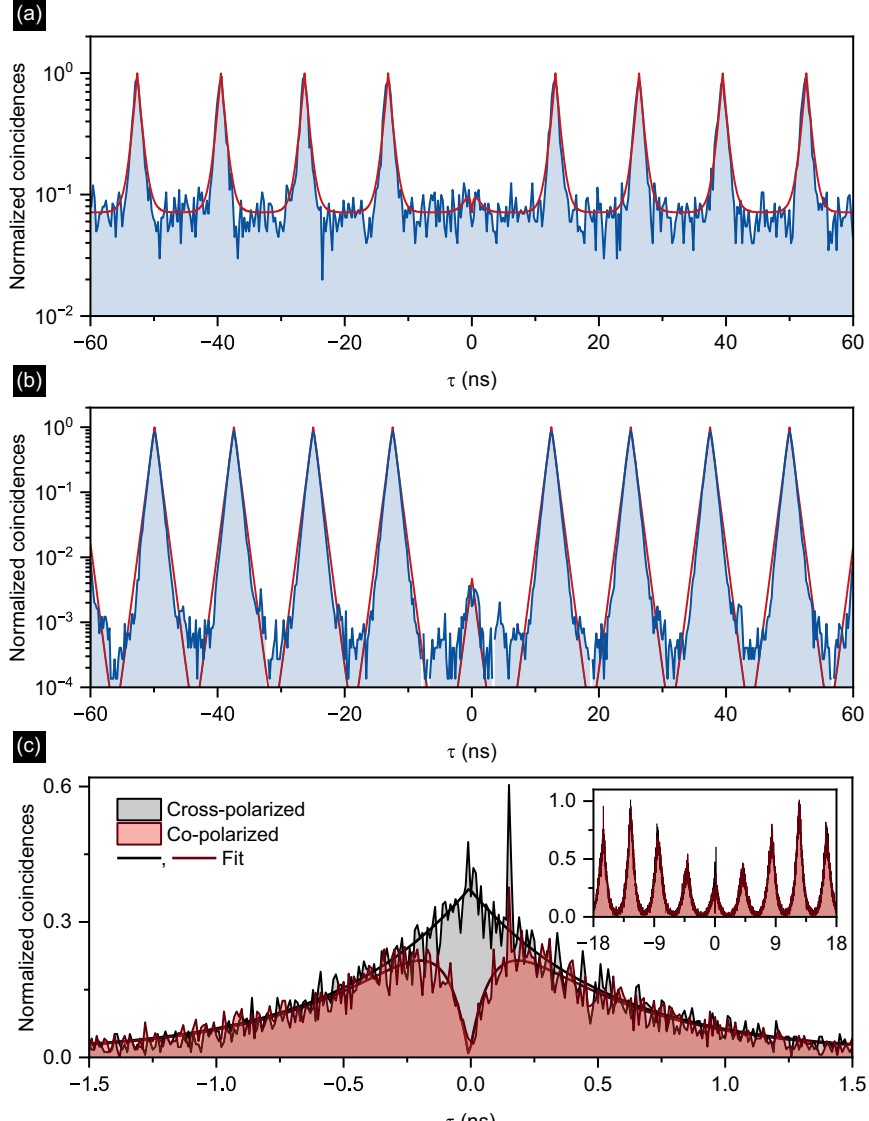

**Fig. 4 | Quantum optical experiments on device #2. a, b** The second-order autocorrelation function $g^{(2)}(\tau)$ of the triggered photons for the (**a**) above-band excitation, and (**b**) LO-phonon-assisted quasi-resonant excitation. **c** Hong-Ou-Mandel histogram for cross- and co-polarized photons evidencing the indistinguishability by the two-photon interference visibility of $V = (19.3 \pm 2.6)$ %, and the post-selected value of $V_{\mathrm{PS}} = 99.8^{+0.2}_{-2.6}$%. Inset: Data for larger delay $\tau$ range.

favorably with previous reports on non-deterministically fabricated QD-CBGs.

## Photon indistinguishability

Finally, we explore the photon indistinguishability of QD-CBG #2 by Hong-Ou-Mandel (HOM)-type two-photon interference (TPI) experiments[20] (see Methods and Supplementary Note 7 for details on the experimental setup, data analysis, and complementary TPI measurements). The HOM histograms recorded for co- and cross-polarized measurement configurations are presented in Fig. 4c, and were obtained under pulsed quasi-resonant excitation with identical experimental conditions as the $g^{(2)}(\tau)$ measurement presented in the previous section (at $0.04 \times P_{\mathrm{sat}}$). The data shown in Fig. 4c is not corrected for multi-photon events or contributions from residual laser light.

The HOM histograms feature a characteristic pattern that we analyze following the methodology described in ref. 21. The reduced area of the central peak in the co-polarized measurement, as compared to the maximally distinguishable cross-polarized measurement, is a distinct signature of the two-photon coalescence due to a significant

degree of photon indistinguishability. The visibility of the TPI is calculated from the ratio of the fitted central peak area in the co- and cross-polarized measurements as $V = 1 - A_{\mathrm{Co}}/A_{\mathrm{Cross}}$. We obtain a TPI visibility of $V = (19.3 \pm 2.6)\%$ with the accuracy being the propagated fitting errors reflecting the statistics of the experimental data. The temporally post-selected visibility at zero delay time ($\tau = 0$) is $V_{\mathrm{PS}} = 99.8^{+0.2}_{-2.6}\%$, limited only by the system temporal response.

## Photon coherence time

From the width of the central dip, we extract a photon coherence time of $T_2 = (103 \pm 13)$ ps, while the highest extracted coherence time is $T_2 = (176 \pm 9)$ ps, measured for the lowest excitation power (see Supplementary Note 7). Given the Purcell-reduced lifetime of $T_1 = 400$ ps, this results in a $T_2/T_1$-ratio of 0.44, comparing favorably with previous reports for QDs emitting in the telecom C-band[22,23]. The still relatively short coherence time observed in our work is mainly attributed to fluctuating charges in the QD environment, suggested by the observed time-dependent spectral diffusion of emission lines, which most probably limits the observed indistinguishability. The coherence

properties may be further improved by implementing electrical charge stabilization via electric gates or using droplet epitaxy as an alternative growth technique[22].

To gain further insights into the coherence properties, we performed direct measurements of the $T_2$ time using an all-fiber-based Michelson interferometer (MI; see Methods). We extract a coherence time of up to $(62 \pm 3)$ ps for the QD-CBG #2 under weak CW above-band excitation (see Supplementary Note 7). Note that while this value is lower than the $T_2$ time extracted from the dip in the HOM experiment in Fig. 4c, a direct comparison is not possible due to the different excitation schemes applied. An analysis of MI data from a total of three different CBG devices yields $T_2$ values between 18 ps and 60 ps. These numbers compare favorably with MI-measured coherence times of 6–30 ps obtained for three different QDs in planar regions on the same sample. The observed coherence times are comparable with reports in the literature for SK InAs/InP QDs[22]. While the direct comparison of the $T_2$ values measured for QDs with and without CBG should be treated with care due to the relatively low statistics, these results indicate that the microcavity integration does not degrade the optical coherence of the emitted photons. Future work in this direction may include a more elaborate study allowing for deeper insights into the limiting dephasing mechanisms and their timescales, e.g., by applying photon-correlation Fourier spectroscopy[24].

## Discussion

The scalable fabrication of active quantum photonic devices operating in the telecom C-band has been a long-standing challenge. This is mainly due to the random size and strain distribution of epitaxially grown QDs causing an inhomogeneous broadening of the emission and difficulties in localizing suitable QDs due to the high electronic noise level of detector arrays sensitive around 1.55 μm wavelength. In this work, we present a solution to this problem based on a hybrid sample design. We fabricate an InP layer containing epitaxial QDs on top of a Al reflector placed on a Si wafer carrier. This geometry significantly enhances the photon extraction efficiency from the QDs by a factor > 7 enabling the localization of single QDs in a wide-field imaging setup with a thermo-electrically cooled InGaAs camera. For the 10% brightest QDs our setup achieves an imaging SNR > 15.5 and a localization uncertainty of ~80 nm with respect to alignment marks. After final EBL processing, we achieve an overall uncertainty of ~90 nm for fabricating a nanophotonic device around pre-selected QDs. The localization accuracy of our setup is comparable to setups operating in the 770-950 nm range (48 nm[25], 30 nm[26]) where Si-based sensors with four orders of magnitude lower electronic noise can be used. The accuracy in our setup can be further improved by using higher-NA objectives inside the cryostat together with an overall increased microscope magnification, which was reported to reduce the localization accuracy down to 5 nm[27]. Alternatively, in-situ EBL[28] or photolithography[29] scanning techniques provide a similar accuracy down to ~35 nm, but are comparably slow, require cathodoluminescence signal for QD localization or, in case of photolithography, are not suitable to define reliably sub-μm features.

To exemplify our approach, we fabricate CBG cavities with a resonance wavelength of 1.55 μm around some of the pre-selected QDs. The low QD density of $3.1 \times 10^8/cm^2$ guarantees their spatial isolation sufficient for the imaging procedure while maximizing the number of devices that can be fabricated per field. The QD-to-CBG coupling is evidenced by a Purcell factor of $F_P = (5.0 \pm 0.4)$, further increasing the single-photon emission rate and final source brightness. Using our approach, we obtain a total process yield of 30% for finding a pre-selected QD spectrally matching the CBG cavity, which is a significant improvement compared to the yield achievable with a random placement approach (~0.7%). For our QD-CBG device, we measure a photon extraction efficiency of $(16.6 \pm 2.7)\%$ with a NA = 0.4 objective,

which is comparable to previously reported devices fabricated probabilistically and operating at C-band wavelengths[9], as well as deterministically fabricated CBGs at O-band with In(Ga)As/GaAs QDs[30]. However, no cavity positioning accuracy, fabrication yield, and HOM visibility data were provided in ref. 30. The discrepancy between the simulated and measured photon extraction efficiency is attributed to the residual defects in the epitaxial material and possible material damage due to the dry etching. The fabrication can be optimized to eliminate both effects.

Our QD-CBG devices feature excellent single-photon emission purities with raw values down to $g^{(2)}(0) = (3.2 \pm 0.6) \times 10^{-3}$, beating previous records for non-deterministically[9] and deterministically[30] fabricated QD-CBGs, as well as most QDs operating in the C-band[31,32], while being not yet competitive with the state-of-the-art[33].

Importantly, we report triggered TPI experiments for InP-based cavity-coupled QDs with emission wavelengths in the telecom C-band, which is crucial for applications in quantum information processing (QIP). We generate indistinguishable photons with a TPI visibility up to $V = (19.3 \pm 2.6)\%$ and a post-selected value of $V_{PS} = 99.8^{+0.2}_{-2.6}\%$ at zero time delay. Previous reports on C-band QD-SPSs by other groups were based either on droplet epitaxy InAs/InP QDs in planar structures[22,34] or InAs QDs grown on GaAs followed by an InGaAs metamorphic buffer, also located in planar structures[23,35] or embedded in randomly placed CBGs[9]. The photon coherence time measured using a Michelson interferometer is up to $(62 \pm 3)$ ps for the QD-CBG #2 under weak CW above-band excitation.

Further improvement in the photon indistinguishability is of utmost importance for applications in QIP. This is challenged by the strong coupling of QDs to their semiconductor environment via charge and spin noise, both causing QD decoherence[21,36]. It is thus important to stabilize the QD environment by removing the excess charge carriers from the vicinity of QDs, e.g., by integrating them into a p-i-n junction, which is expected to increase the photon coherence and indistinguishability substantially. On the other hand, tuning the QD emission energy using strain[37,38], or quantum-confined Stark effect[39,40], would address the challenge of QD ensemble inhomogeneous broadening by fine-tuning the QD energy to match the cavity mode. The QD tuning is feasible using the reported approach but requires a different cavity design[41,42]. Implementing coherent optical pumping schemes, such as two-photon resonant excitation[43], also for scalably fabricated devices, while avoiding the excess charge carriers that could originate, e.g., from the electrical QD excitation, is a crucial next step to further improve the photon coherence time and hence indistinguishability[44].

Moreover, the InP material system used in our work appears to be advantageous for QD-based quantum photonic devices operating in the C-band and compared to GaAs-based devices. Despite the careful strain engineering involved in the epitaxy of QDs on GaAs[45], the metamorphic buffer complicates the device engineering and QD growth. In contrast, an unstrained InP system is free from threading dislocations that would be a source of dangling bonds causing non-radiative recombination, thus lowering the efficiency[46].

In conclusion, our work opens the route for the high-throughput fabrication of telecom C-band wavelength quantum photonic devices with QDs delivering flying qubits, i.e., single or entangled photons[47], or acting as a non-linear element for QIP[48]. Improvements in our optical imaging setup will further increase the device yield and positioning accuracy, while the electric control and coherent excitation of QD emitters will further push the achievable photon indistinguishability.

## Methods

### Epitaxial growth and fabrication of planar structure with QDs
The structures were grown on epi-ready (001)-oriented InP substrates by the low-pressure metalorganic vapor-phase epitaxy (MOVPE)

TurboDisc® reactor using arsine (AsH$_3$), phosphine (PH$_3$), tertiarybutylphosphine (TBP), trimethylgallium (TMG) and trimethylindium (TMIn) precursors with H$_2$ as a carrier gas. We grow the 0.5 μm-thick InP buffer followed by 200 nm-thick In$_{0.53}$Ga$_{0.47}$As lattice-matched to InP etch-stop layer and a 156 nm-thick InP layer at 610 °C. Then, the temperature is decreased to 493 °C, stabilized under TBP for 180 s and AsH$_3$ for 27 s. The nucleation of QDs occurs in the near-critical regime of Stranski-Krastanov growth mode after deposition of nominally 1.22 ML-thick InAs at growth rate 0.53 ML/s under TMIn and AsH$_3$ flow rates of 11.8 μmol/ min and 590 μmol/ min, respectively (V/III ratio of 50). Nucleated QDs are annealed for 3.5 s at the growth temperature in AsH$_3$ ambient before the deposition of a 156 nm-thick InP capping layer (12 nm at 493 °C, and the remaining 144 nm after increasing the temperature up to 610 °C) what finishes the growth sequence.

### Fabrication of the sample for μPL imaging

After the QD epitaxy, SiO$_2$ is deposited in plasma-enhanced chemical vapor deposition (PECVD). This layer is intended to be 358.6 nm-thick, and it is covered with a 120 nm-thick Al layer deposited via electron-beam evaporation. The flipped structure is bonded to the Si chip carrier utilizing spin-coated AP3000 adhesion promoter and benzocyclobutene (BCB) on Si and AP3000 on the InP wafer. The bonding is done by applying the force of ~2 kN in vacuum at 250 °C. The substrate removal step is done by ~ 60 min dip in HCl and the InGaAs etch-stop layer is subsequently removed in H$_2$SO$_4$:H$_2$O$_2$:H$_2$O = 1:8:80 mixture. Next, by employing electron-beam lithography (EBL) followed by inductively coupled plasma-reactive ion etching (ICP-RIE) to etch InP down to SiO$_2$, the square imaging fields were fabricated with 50 μm side-length and AMs dedicated to EBL outside the fields. Therefore, there are different AMs for optical imaging (edges of the imaging fields) and for EBL alignment marks detection (InP crosses). This approach is justified by the simplification of the fabrication flow by avoiding the deposition of metallic AMs, relying instead on the outline of the field visible due to the μPL signal scattering from its edges. The material contrast between InP and SiO$_2$/Al regions is sufficient for the AM detection during the EBL alignment step.

### Modeling of the CBG

The QD is modeled as a classical dipole[49], and the numerical simulations of the CBG geometry are performed using a modal method employing a true open boundary condition[50]. See the Supplementary Note 1 for further details.

### Deterministic fabrication of QD-CBG devices

After determining the positions of QDs with respect to the AMs, the CBG pattern is defined in the CSAR e-beam resist in e-beam lithography using high-precision alignment based on the InP mark detection in JEOL JBX-9500FSZ e-beam writer. The pattern is transferred into the PECVD-deposited 110 nm-thick SiN$_x$ layer using ICP-RIE with SF$_6$-based etch recipe. Residual CSAR is stripped in Remover 1165 followed by 10 min descum in the barrel-type plasma asher. Subsequently, the pattern is transferred into the InP layer in ICP-RIE by HBr-based etch. The calculated design is first scaled and fabricated using a nominally identical heterostructure to investigate the mode energy vs. size dependence and to account for the fabrication imperfections. Additionally, we experimentally determine the ~15 nm temperature-induced blueshift of the mode energy between a room and low temperature resulting from the contraction of the structure (introducing size and strain changes), as well as from the change of the refractive indices of the layers.

### Optical characterization of devices

The structure with QD-CBG devices is held in a helium-flow cryostat allowing for control of the sample temperature in the range of 4.2 − 300 K. For our standard μPL studies, the structures are optically excited through a microscope objective with NA = 0.4 or 0.65 and 20 × or 50 × magnification using 660 nm or 805 nm light generated with semiconductor laser diodes. The same objective is used to collect the μPL signal and to direct it for spectral analysis into a 1 m-focal-length monochromator equipped with a liquid-nitrogen-cooled InGaAs multichannel array detector, providing spatial and spectral resolution of ~2 μm and ~25 μeV, respectively.

The photon extraction efficiency and time-resolved μPL are measured in the same setup. Here, QDs are excited by ~50 ps-long pulses with a repetition rate of 80 MHz and a central wavelength of 805 nm. At the same time, the second monochromator output port is equipped with the fiber coupling system, transmitting the signal to an NbN-based SNSPD (Scontel) with ~87% quantum efficiency in the range 1.5 − 1.6 μm and ~200 dark counts per second. A multichannel picosecond event timer (PicoHarp 300 by PicoQuant GmbH) analyzes the single photon counts as a time-to-amplitude converter. The overall time resolution of the setup is ~80 ps. Experimental setups are shown in Supplementary Note 4, and data on the setup transmission efficiency used for determining the photon extraction efficiency is given in Supplementary Note 5.

### Photon autocorrelation measurements

For the photon-autocorrelation measurements, QD-CBG devices were optically excited using a Ti:Sapphire (Ti:Sa) laser (Coherent Mira-HP) or a widely tunable ps-pulsed laser system based on an optical parametric oscillator (OPO) (picoEmerald by APE GmbH) with repetition rates of 76 MHz and 80 MHz, respectively. We use a fiber-coupled bandpass filter (FWHM = ~ 0.4 nm) for spectrally selecting the QD emission, followed by a 50:50 fiber beam splitter. For the off-resonant excitation, we use a microscope objective with NA = 0.7 and 100 × magnification and excite the QD emission with ~2 ps-long pulses at 830 nm from the Ti:Sa. The signal is detected with a pair of SNSPDs with ~87% and ~92% quantum efficiency at 1550 nm. For the quasi-resonant excitation, we use an aspheric lens (NA = 0.6) mounted inside a low-vibration closed-cycle cryostat (attoDRY800 by Attocube Systems AG) cooled to 4.5 K. Here, the OPO-laser is used and adjusted to a pulse length of 5 ps. Single photons are detected via SNSPDs with 80% detection efficiency at 1550 nm and 57 ps timing jitter (complete system temporal response). The excitation energy was determined in photoluminescence excitation experiments to be 0.83537 eV (37.57 meV above the QD emission energy, cf. Supplementary Note 7), which was also used for following TPI experiments.

### Photon-indistinguishability measurements

In the TPI experiments, an additional 4 ns delay was introduced between consecutive laser pulses by adding an imbalanced free-space Mach-Zehnder interferometer (MZI) in the excitation path, which was compensated in the HOM setup on the detection side. Hence, the excitation sequence is composed of pairs of pulses separated by 4 ns, every 12.5 ns corresponding to 80 MHz laser repetition rate. Free-space waveplates were used to match the polarization for the TPI inside the fiber beam splitter. The exact polarization was set by using a polarimeter at the beam splitter output in combination with a laser tuned to the QD emission wavelength. Fine-tuning the relative delay between both MZI arms was used to match precisely the detection and excitation delay, respectively. The contrast of classical Michelson interference of the laser with itself was used for optimization. See Supplementary Note 7 for the details of the HOM data analysis and the scheme of the experimental setup.

### Coherence measurements

For measurements of the $T_2$ time, an all-fiber-based Michelson interferometer (MI) was implemented[22], consisting of a 2 × 2-port 50 : 50 fiber beam splitter with both exit ports terminated by a

Faraday mirror, reflecting the light with 90° polarization rotation. The necessary coarse and fine temporal delay is controlled by a variable optical delay stage and a piezo-driven fiber stretcher in the two MI arms, respectively. The single-photon signal is coupled to one input port of the MI and detected at the second input port using a SNSPD. The MI setup in its configuration features 80% overall transmission (excluding the BS) and allows for the measurement of coherence times of up to 1 ns. The maximally achievable interference contrast was measured with a CW laser at 1550 nm to be 98%, limited only by the slight intensity mismatch due to the reduced transmission through the optical delay line. For each temporal delay adjusted via the coarse variable delay line, a fine temporal scan is performed via the fiber stretcher, resulting in interference fringes with an amplitude depending on the overall delay. The interference fringes are evaluated by subtracting a constant amount of dark counts and evaluating the interference contrast via $v = (I_{max} - I_{min})/(I_{max} + I_{min})$. Finally, the $T_2$ time is extracted by fitting a two-sided exponential decay to the interference visibility $v$ data as a function of the coarse delay set in the MI with the uncertainty representing the fit accuracy.

## Data availability

The source data for plots in Figs. 1–4 and 50 representative $\mu$PL maps with full results of data analysis leading to QD localization have been deposited in the Figshare database under accession code[51].

## Code availability

The codes of this study are available from the corresponding authors upon reasonable request.

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

## Acknowledgements

The authors acknowledge financial support from the Danish National Research Foundation through NanoPhoton - Center for Nanophotonics, grant number DNRF147, and the Center for Macroscopic Quantum States (bigQ), grant number DNRF142. P.H., M.G.M., P.M., A.M. and B.K. acknowledge financial support from the Polish National Science Center (Grants No. 2020/36/T/ST5/00511, 2020/39/D/ST5/02952, 2020/39/D/ST5/03359). D.A.V. and T.H. acknowledge financial support by the German Federal Ministry of Education and Research (BMBF) via the project "QuSecure" (Grant No. 13N14876) within the funding program Photonic Research Germany, the BMBF joint project "tubLAN Q.0" (Grant No. 16KISQ087K), and by the Einstein Foundation via the Einstein Research Unit "Quantum Devices". N.G. acknowledges support from the European Research Council (ERC-CoG "UNITY", Grant No. 865230), and from the Independent Research Fund Denmark (Grant No. DFF-9041-00046B).

## Author contributions

E.S. conceived the project. M.S. provided the concept of optical imaging and designed spectroscopic experiments. P.H. and E.S. optimized and performed the QD growth. P.M., P.H., B.K. and M.S. constructed the μPL imaging system. B.G. and N.G. designed the CBG. P.H. with the help of A.S., M.X. and K.Y. developed the nanofabrication process. P.H. and E.Z.-O., with the help of M.G.M. and P.M. and the advice of M.S. carried out imaging experiments and device characterization, including above-band excitation photon autocorrelation. P.H. localized the QDs and fabricated the devices. D.A.V. performed the photon autocorrelation and two-photon interference experiments with the help of M.W., E.Z.-O., A.M., and under the supervision of T.H. P.H., E.Z.-O., D.A.V., M.W., A.H., with the advice of A.M., M.S. and T.H. analyzed and interpreted the data. P.H., with the help of A.H. and D.A.V., and with the advice of T.H., M.S., and E.S. wrote the manuscript and SI.

## Competing interests

The authors declare no competing interests.
