## [Peer Review File · Nature Communications]

High-throughput quantum photonic devices emitting indistinguishable photons in the telecom C-bandEditorial Note: Parts of this Peer Review File have been redacted as indicated to remove third-party material where no permission to publish could be obtained.

REVIEWER COMMENTS

Reviewer #1 (Remarks to the Author):

The manuscript by Holewa et al., demonstrates a technique for deterministic fabrication of nanophotonic devices containing single epitaxial InP/InAs quantum dots, capable of providing C-band telecom single-photon emission. The fabrication method employs a wide-field single QD micro-photoluminescence imaging technique which allows in principle a high throughput for the localization of single QDs, in comparison e.g., with confocal scanning microscopy. Crucially, the use of wide-field imaging of QDs at telecom wavelengths is shown to be possible with a TE-cooled camera. This was made possible by a relatively high QD light extraction efficiency from the semiconductor, achieved in samples where the QD-hosting InP film is placed above a metallic mirror via an adhesive wafer bonding method. While fabrication of such hybrid, bonded samples had been demonstrated by the authors in a prior publication, here one of the main innovations is in the effectiveness for imaging and deterministic QD device integration. The authors demonstrate the effectiveness of the technique by fabricating circular Bragg grating cavities containing single positioned QDs.

While a lot of the device work is directly based upon prior techniques developed by other groups, the QD material system is very interesting because of the ability to reach C-band telecom wavelengths, and the ability to perform wide-field deterministic positioning demonstrated here certainly makes the platform very promising as a whole. I also appreciate the pulsed two-photon interference measurements, to my knowledge the first reported for this class of QDs, and in cavity-coupled photons, which provide critical information about the reasonable promise and challenges of the platform, in particular when compared to alternative, perhaps more mainstream material systems.

I think the manuscript could be suitable for Nature Communications provided that the following comments are properly addressed:

1 - Wide-field imaging of telecom O-band QDs for deterministic fabrication was demonstrated in the past by Hu et al. , *Photon. Res.* 10 B1 (2022) . I think this work should be cited and differentiated from what was done here. My understanding is that in that publication a LN2 cooled InGaAs camera was used, which provides a somewhat lower noise background. Also, have the authors attempted to perform imaging in samples that did not feature the back metallic mirror? I think the results of such experiments should be commented on - were the QDs visible at all, and at what SNR? More generally, though only optionally, I think it would be extremely helpful if the authors could expand on what the achievable SNR is as a function of extraction efficiency, given typical detector noise values.

2 - It seems the maximum experimental collection efficiency of 16.6 % is considerably lower than the expected from simulations. The authors should elaborate more on why this is so. In particular, QD positioning has a strong impact on the extraction efficiency. Given that the estimated QD positioning uncertainties (~ 100 nm) are somewhat smaller than the overall QD position range, given in Fig. S2, within which the efficiency should remain above 50 %, it is somewhat surprising to see such low extraction.

4 - Can the authors comment on whether the low coherence times are due to spectral diffusion or pure dephasing? Also, is there any evidence that fabrication or growth have major contributions to the low coherence times? In particular, even though coherence times are not so good, they are more or less comparable with those of other telecom QDs in fabricated nanostructures. Considering that the present QDs are grown by MOVPE, which is not a preferred method for this type of work, I suggest that the authors expand upon this discussion.

5 - Regarding the lower purity and coherence times at higher powers, is the presence of other nearby QDs involved? While the devices feature single QDs emitting at a narrow filter window, there is still a relatively high density of QDs surrounding the positioned ones, with emission elsewhere, so it's not unreasonable to think that quasi-resonant excitation could be accessing such QDs. I suppose in this case potentially growing lower densities of QDs could help - a related question, then, is whether lower QD densities would be a possibility.

6 - Relatedly, have the authors attempted LA-photon or direct resonant excitation of the QD? It would be interesting to see if the coherence time can be significantly improved, since these two excitation methods are minimally detrimental to the single-photon coherence times.

7 - What is limiting the post-selected visibilities for the different excitation powers, or why is there such large variability? Close to zero delay, well below the coherence time, it should be always very high, unless there's a significant spatio-temporal and polarization mismatch between the two photons being interfered. I think polarization and spatial mismatch can be well controlled in the experiment, though polarization control can be tricky when using optical fibers. Is temporal mismatch, e.g. due to jitter caused by quasi-resonant excitation, an issue?

8 - The authors should further clarify what QDs were selected as references for the radiative rate enhancement estimates. E.g., were the 8 reference QDs located inside cavities, but spectrally detuned? Or were they completely outside any cavities? This is not very clear in the text, though it's somewhat important because there's a chance for radiative rate suppression of QDs located at resonance nodes in the cavity.

Reviewer #2 (Remarks to the Author):

In this experimental research paper, the authors presented the generation of single, indistinguishable photons emitted by localized InAs/InP QDs in the telecom C-band range. To date, a truly scalable technology for generating on-demand, high repetition rate single-photons at 1550 nm has remained elusive. This is despite the high relevance of the telecom C-band for quantum networks mediated by flying qubits.

In this report, the authors demonstrated deterministic coupling of pre-selected quantum emitters with circular Bragg gratings (CBG). These devices generate single photons with high purity and indistinguishability. The data presented were acquired using suitable methods, well analyzed, interpreted, and presented with good detail, ensuring their technical

accuracy. The authors presented compelling evidence to support their assertions regarding high photon indistinguishability, reliable positioning, and Purcell effect. All necessary controls have been duly incorporated in the study.

However, as elaborated below, the results are mainly incremental with low potential to advance the field. As such, publication in Nature Communications is NOT recommended.

1) The demonstrated positioning involves CBG, which possesses a quite large mode volume (V) and a low-quality factor (Q) (fig 3a). While this is useful for showcasing deterministic coupling, the most crucial quantum devices necessitate an ultra-high Q/V ratio. This is evident also in the low Purcell factor reported.

2) This work primarily builds upon the existing literature and previous publications from the same group. (i) The positioning of QDs to photonic resonators (such as CBG) has been demonstrated multiple times using the same PL technique. (ii) Besides the SPS (already showed by several groups and in ref. 10 by the authors), this paper has not demonstrated the versatility of this structure in enabling new quantum devices. For example, it is unclear if the CBG can be tuned independently from the QDs or vice versa. Additionally, questions remain regarding the possible implementation of quantum optics schemes, such as a quantum repeater, or the ability to voltage-control the QD exciton or trigger emission. In this regard it is not a breakthrough that significantly extends the knowledge and fosters innovation, ultimately contributing to the development of new technologies, quantum protocols, or device research.

3) The authors gave a certain emphasis (title) to the scalability of their approach. Scalability requires a clear context for better understanding and accurate interpretation. Without such knowledge, determining the scalability of the structure remains ambiguous.

Here, it is unclear if the structure needed to optimize the localization (Fig. 1a and fig. 2b) is tolerant with the implementation of real quantum devices in terms of scalable integration and manufacturing. Scalability in terms of integrating other functional photonic devices would rely on the compatibility of structures and materials. In this regard, the authors should show how that QD-CBG structure in presence of metal markers can be coupled to a

heterogeneous nanophotonic structure. Thorough analysis and testing should be conducted to ensure seamless interaction and optimized performance.

Scalability in terms of manufacturing refers to the ability to produce large volumes of these photonic devices while maintaining consistent quality. However, self-assembled QDs nucleate on the surface in random location and with different exciton emission spread on the spectrum. Thus, the QD-CBG structure will emit photon of different wavelengths with different exciton configuration and properties. The presented approach must go through a non-parallelizable sorting of the QDs.

Reviewer #3 (Remarks to the Author):

In the manuscript entitled “Scalable quantum photonic devices emitting indistinguishable photons in the telecom C-band” by Holewa et al. the authors report on a quantum dot-based source of single photons. The source can emit photons at $\sim 1550\text{nm}$ and the collection efficiency is enhanced using Circular Bragg Grating (CBG) cavity.

The central point of the contribution relates to scalability. Namely, many different photonic structures can be fabricated to enable higher collection efficiency or Purcell factor.

However, depending on the complexity of the fabrication the yield of functioning structures is commonly quite low. Furthermore, most of the structures need what is commonly known as deterministic fabrication – the photonic structure needs to be accurately placed over the site of formation of the emitter, otherwise the collection efficiency and Purcell effect will not be achieved.

The CBG cavities are photonic structures known to be notoriously hard to fabricate deterministically, which in return leads to reduced value of the Purcell enhancement compared to the design value. The most complicated element of the CBG implementation is elimination of birefringence that is known to reduce the applicability of the device – one cannot generate entangled photon pairs nor get a very good value of the two-photon interference.

While this was shown for the devices operating in wavelength range 750-950, a deterministic CBG device was never shown in telecom range. One of the reasons for this is that the emitter imaging in the wavelength range is very problematic. The imaging cameras are conceptually different – the pixel size is larger, and noise is stronger, which makes the

imaging quite demanding. However, the authors have overcome this problem and reached the placement accuracy of $\sim 90\text{nm}$.

Having summarized the presented results; I must conclude that submitted work might belong to portfolio of Nature Communications however, some issues need to be correctly addressed:

1. Could you please estimate the improvement needed in imaging to reach the accuracy where the collection efficiency and Purcell enhancement start to be more like what is predicted in the device theoretical simulation?
2. Can you provide the lifetime fit made using linear scale. The background in the reference dot signal looks high enough to alter the log-fit result and artificially extend the lifetime. This could lead to an erroneous estimate of the Purcell enhancement.
3. The two-photon interference is not very high, and this is attributed to the magnetic and electrical field fluctuations. What about the birefringence? Can the authors comment on the birefringence induced by the structure?

Response to Reviewers' comments

We sincerely thank the Reviewers for their time and efforts, and we greatly appreciate their recognition of our work and constructive comments. Below, we give a detailed response to their reports. The changes made to the manuscript and supplementary information are indicated in green.

Reviewer #1 (Remarks to the Author):

The manuscript by Holewa et al., demonstrates a technique for deterministic fabrication of nanophotonic devices containing single epitaxial InP/InAs quantum dots, capable of providing C-band telecom single-photon emission. The fabrication method employs a wide-field single QD micro-photoluminescence imaging technique which allows in principle a high throughput for the localization of single QDs, in comparison e.g., with confocal scanning microscopy. Crucially, the use of wide-field imaging of QDs at telecom wavelengths is shown to be possible with a TE-cooled camera. This was made possible by a relatively high QD light extraction efficiency from the semiconductor, achieved in samples where the QD-hosting InP film is placed above a metallic mirror via an adhesive wafer bonding method. While fabrication of such hybrid, bonded samples had been demonstrated by the authors in a prior publication, here one of the main innovations is in the effectiveness for imaging and deterministic QD device integration. The authors demonstrate the effectiveness of the technique by fabricating circular Bragg grating cavities containing single positioned QDs.

While a lot of the device work is directly based upon prior techniques developed by other groups, the QD material system is very interesting because of the ability to reach C-band telecom wavelengths, and the ability to perform wide-field deterministic positioning demonstrated here certainly makes the platform very promising as a whole. I also appreciate the pulsed two-photon interference measurements, to my knowledge the first reported for this class of QDs, and in cavity-coupled photons, which provide critical information about the reasonable promise and challenges of the platform, in particular when compared to alternative, perhaps more mainstream material systems.

We would like to thank the Reviewer for the overall positive assessment of our work and for emphasizing its relevance to the photonics community.

I think the manuscript could be suitable for Nature Communications provided that the following comments are properly addressed:

Editorial Note: Column 2 of Table R1 has been redacted.

1 - Wide-field imaging of telecom O-band QDs for deterministic fabrication was demonstrated in the past by Hu et al., Photon. Res. 10 B1 (2022). I think this work should be cited and differentiated from what was done here.

The recently published work by Shi-Wen Xu *et al.*, Photon. Res. 10, 8 (2022) (added Ref. [29]) reports on the PL imaging of emission from In(Ga)As/GaAs at 1320 nm, followed by the fabrication and characterization of hybrid CBGs. While the work by Xu *et al.* misses important details on the imaging procedure and HOM results (see further discussions below), we agree with the Reviewer that we should cite this work in our article.

The properties and parameters of Xu *et al.* and our work are summarized in Table R1, with the main points highlighted in bold.

Table R1. Comparison of the experimental conditions and results presented in Shi-Wen Xu *et al.*, Photon. Res. 10, 8 (2022) and in our work.

	Shi-Wen Xu et al. , Photon. Res. 10, 8 (2022) - Ref. [29]	Our work
Wavelength range		1550 nm, C-band
Material system		InAs/InP
Cavity positioning accuracy		90 nm
HOM visibility		(19.3 ± 2.6) %
$g^{(2)}(0)$ for non-resonant pulsed excitation under high excitation		$g^{(2)}(0) = (0.05 \pm 0.02)$ at 50% saturation power (Fig. 4a)
Epitaxial method		metalorganic vapour-phase epitaxy
Photon extraction efficiency		(16.6 ± 2.7) %
Fabrication yield		30 %
Device footprint		7.7 μm
Purcell factor		5 ± 0.4
Detector used		NIRvana 640

The main qualitative difference between both works is the promise they bring to the photonic community—the authors of Shi-Wen Xu *et al.*, Photon. Res. 10, 8 (2022) do not provide any details on the localization of QDs (neither positioning accuracy nor fabrication yield is given).

¹ Fig. 4f, at 3.8 μW

Therefore, although the fabrication of a CBG working in the O-band was reported, there are no details regarding the approach's scalability nor how many devices were successfully fabricated.

Besides, the works differ qualitatively concerning the material system (In(Ga)As/GaAs vs. InAs/InP) and operation wavelength (1320 nm vs. 1550 nm), both providing different motivations and perspectives for the respective platforms.

We have summarized the differences in the *Discussion* section of our article:

For our QD-CBG device, we measure a photon extraction efficiency of $(16.6 \pm 2.7) \%$ with a $NA = 0.4$ objective, which is comparable to previously reported devices fabricated probabilistically and operating at C-band wavelengths⁹, as well as deterministically fabricated CBGs in the O-band with In(Ga)As/GaAs QDs²⁹. However, in contrast to our work, no cavity positioning accuracy, fabrication yield, and HOM visibility data were provided in Ref. [29].

Our QD-CBG devices feature excellent single-photon emission purities with raw values down to $g^{(2)}(0) = (3.2 \pm 0.6) \times 10^{-3}$, beating previous records for non-deterministically⁹ and deterministically²⁹ fabricated QD-CBGs (...)

[9] C. Nawrath, R. Joos, S. Kolatschek, S. Bauer, P. Pruy, F. Hornung, J. Fischer, J. Huang, P. Vijayan, R. Sittig, M. Jetter, S. L. Portalupi, and P. Michler, "Bright Source of Purcell-Enhanced, Triggered, Single Photons in the Telecom C-Band," *Adv. Quantum Technol.*, 2300111 (2023).

[29] S.-W. Xu, Y.-M. Wei, R.-B. Su, X.-S. Li, P.-N. Huang, S.-F. Liu, X.-Y. Huang, Y. Yu, J. Liu, and X.-H. Wang, "Bright single-photon sources in the telecom band by deterministically coupling single quantum dots to a hybrid circular Bragg resonator," *Photon. Res.* 10, B1–B6 (2022).

(continued) My understanding is that in that publication a LN2 cooled InGaAs camera was used, which provides a somewhat lower noise background.

The Reviewer is correct in stating that the NIRvana LN detector used by Xu *et al.* has a lower noise floor compared to the NIRvana 640 detector we used in our experiment. Importantly, in the range of 1550 nm, the quantum efficiency of the NIRvana 640 detector is constant, around 83%, while the efficiency of the NIRvana LN in this wavelength range is strongly wavelength-dependent. Therefore, we choose the 640 detector to obtain μ PL images and localize QDs.

In the following, we summarize technical details on Quantum efficiency and noise of these two detector arrays.

Quantum efficiency

The spectrally dependent quantum efficiency of the NIRvana LN and the NIRvana 640 is summarized in Fig. R1. The NIRvana LN detector reaches 48% at 1550 nm, while the value rapidly decreases between 1520 nm (85%) and 1580 nm (~2%). In contrast, the efficiency of the NIRvana 640 has an absorption edge at ~1650 nm, and the efficiency has only a weak spectral dependence in the range of interest at 1500-1600 nm with an average efficiency above 80%.

FIGURE REDACTED

Fig. R1. A comparison between the quantum efficiency for (a) the liquid nitrogen-cooled detector NIRvana LN applied in Shi-Wen Xu et al., *Photon. Res.* 10, 8 (2022) and (b) NIRvana 640, used in our work, with the efficiency level at 1550 nm additionally marked. Adapted from Teledyne Princeton Instrument's website, source for a: link, source for b: link.²

Noise

The primary source of noise in InGaAs detectors is dark current. The figure of merit is the signal-to-noise ratio (SNR) evaluated considering dark current, pixel size, quantum efficiency, and readout noise.

In Fig. R2, we show the SNR calculated for a signal level comparable to that achieved in our experiment using the online data by Teledyne Princeton Instruments. The vertical dashed line indicates the 2 s exposure time applied in our experiment, which is a tradeoff between SNR and stability of the imaging systems. It can be seen that for the nitrogen-cooled detector, SNR reaches ~ 20 , while for NIRvana 640, SNR is ~ 10 .³

² All hyperlinks given in this document are accessible on the day of submission (November 11th, 2023).

³ The value is different from those reported by us in Supplementary Fig. S5, as we have binned rows to increase the SNR while the producer provides values per μm^2 of the detector's pixel.

FIGURE REDACTED

Fig. R2. A comparison of SNR as a function of exposure time at the signal level comparable to that in recorded μ PL maps. Adapted from Teledyne Princeton Instrument's website, source: link.

The dark current of both detectors was obtained from their data sheets available online and summarized in Table R2. The levels are comparable ($< 10 \text{ e}^-/\text{p/s}$ vs. $< 40 \text{ e}^-/\text{p/s}$) when seen in comparison to the Si EMCCD ($0.02 \text{ e}^-/\text{p/s}$ at maximum) with a four orders of magnitude lower value.

Table R2. Comparison between the figures of merit for NIRvana LN and NIRvana 640 detectors.

TABLE REDACTED

In conclusion, for imaging in the C-band as done in our work, the NIRvana 640 detector appears as the better choice due to the nearly constant and above 80% quantum efficiency till 1600 nm, although the slightly higher dark current as compared to the NIRvana LN detector.

(continued) Also, have the authors attempted to perform imaging in samples that did not feature the back metallic mirror? I think the results of such experiments should be commented on - were the QDs visible at all, and at what SNR?

We initially tried imaging the chips with QDs without the metallic mirror, but it was impossible to discriminate the QD spots from the noisy background. Therefore, we have concluded in the manuscript that the fabrication of the metallic mirror is indispensable for establishing the deterministic fabrication of the QD-containing photonic devices.

As we indicate in section S-III A in the Supplemental Material, the average signal-to-noise ratio for the sample without a mirror is about 1.5, which we calculate based on the analysis of the recorded μ PL maps for the sample with the mirror and the simulated mirror enhancement factor of about 7 at 1550 nm, which was reported in Ref. [10].

We have added this information to the manuscript:

“Overall, this geometry enhances the QD emission in the out-of-plane direction by a factor >7 as compared to bulk InP samples, reaching a total photon extraction efficiency of $> 10\%$ from a single QD for $NA=0.4$. In fact, this design turned out to be crucial for the imaging step, as the SNR of the QD emission was insufficient for investigated structures without a backside mirror (see Supplementary Note 3).”

(continued) More generally, though only optionally, I think it would be extremely helpful if the authors could expand on what the achievable SNR is as a function of extraction efficiency, given typical detector noise values.

We agree with the Reviewer that this information would be helpful to have. Unfortunately, the extraction efficiency reported in Fig. 3c concerns only the QDs located in the devices, and we have not determined the photon extraction efficiency prior to the fabrication of the devices (*planar* photon extraction efficiency η_p). Therefore, we are missing the connection between the SNR and η_p (critical during the localization phase) that the Reviewer is asking for.

Nonetheless, we can perform an extrapolation of the dependence. From our calculations reported in Ref. [10] (Fig. 3b), the expected planar photon extraction efficiency at 1550 nm is $\eta_p = 7\%$. The remaining question is attributing the calculated $\eta_p = 7\%$ to the detector signal level, given the distribution of SNR shown in Supplementary Fig. S5. We estimate that the brightest 10% of QDs reach SNR = 15.5. However, the (standard) average value is SNR = 10.6. We attribute the spectral mismatch between the QD emission line and the central wavelength of the bandpass filter, as well as the difference in the emission intensity between QDs, as the main reasons for the SNR distribution broadening.

Assuming that the theoretically expected level of 7% matches the SNR = 15.5 level, we can simulate the SNR vs. planar extraction efficiency dependence, shown in Fig. R3.

[10] P. Holewa, A. Sakanas, U. M. Gür, P. Mrowiński, A. Huck, B.-Y. Wang, A. Musiał, K. Yvind, N. Gregersen, M. Syperek, and E. Semenova, "Bright Quantum Dot Single-Photon Emitters at Telecom Bands Heterogeneously Integrated on Si," *ACS Photonics* 9, 2273–2279 (2022).

Fig. R3. Simulated dependence of SNR on the planar photon extraction efficiency η_p .

2 - It seems the maximum experimental collection efficiency of 16.6 % is considerably lower than the expected from simulations. The authors should elaborate more on why this is so. In particular, QD positioning has a strong impact on the extraction efficiency. Given that the estimated QD positioning uncertainties (~ 100 nm) are somewhat smaller than the overall QD position range, given in Fig. S2, within which the efficiency should remain above 50 %, it is somewhat surprising to see such low extraction.

The method for estimating photon extraction efficiency assumes that the QD has an internal quantum efficiency of 100% (see the comment below Supplementary Eq. 17 in Supplementary Note 5). Therefore, the reported value of 16.6 % sets the lower limit of extraction efficiency. As the Reviewer observed, this value considerably deviates from the predicted $>50\%$ level for the range of dipole position < 400 nm in Supplementary Fig. 2a.

This discrepancy can be attributed to the non-radiative recombination channels introduced to the QDs due to structural defects propagating from the InP substrate and/or defect states at the side walls of the CBG central mesa, which are introduced during dry etching. These defects most likely cause additional exciton energy relaxation channels. These are, however, difficult to account for and hence not considered in the model.

We added the information to Supplementary Note 5:

This method assumes unity internal quantum efficiency of QDs ($\eta_{int} = 100\%$), so that the QD photon emission rate equals f_{rep} . It is however difficult to determine experimentally the contribution of non-radiative recombination and hence the real value of η_{int} . As a result, a discrepancy arises between the calculated photon extraction efficiency (Fig. 1b) and measured η . The reason for lowered η_{int} (and hence lowered η) can be attributed to the non-radiative recombination channels introduced to the QDs due to structural defects

propagating from the InP substrate or defect states at the side walls of the CBG central mesa, which are introduced during dry etching. These defects most likely cause additional exciton energy relaxation channels. The assumption of $\eta_{int} = 100\%$ thus sets a lower limit of η due to a possible overestimation of the total number of photons emitted by the QD.

4 - Can the authors comment on whether the low coherence times are due to spectral diffusion or pure dephasing? Also, is there any evidence that fabrication or growth have major contributions to the low coherence times? In particular, even though coherence times are not so good, they are more or less comparable with those of other telecom QDs in fabricated nanostructures. Considering that the present QDs are grown by MOVPE, which is not a preferred method for this type of work, I suggest that the authors expand upon this discussion.

Indeed, the coherence times observed in our work (74-176 ps), which were extracted by fitting the HOM dip, are comparable to those directly measured in the prior reports, e.g., for InAs/InP QDs (Ref. [22]) with average coherence times of $T_2 = 51(29)$ ps at saturation power for self-assembled Stranski-Krastanov QDs grown in MOVPE, so the same as used in our study. In Ref. [22], QDs were placed in the *p-i-n* junction, and no nanostructures were fabricated. The authors also showed that the coherence time is significantly higher ($T_2 = 157(72)$ ps) - for QDs grown via the droplet epitaxy method, which can further improve the characteristics of the InAs/InP material system.

The Purcell-enhanced lifetime of the QD state investigated in our work ($T_1 = 400$ ps), combined with the highest extracted coherence time ($T_2 = 176$ ps) at the lowest excitation power, corresponds to a T_2/T_1 ratio of 0.44. This ratio is higher than the values reported in Ref. [22], where no Purcell enhancement was employed. However, the coherence times of Ref. [22] are directly measured (using a Michelson interferometer) and thus are more representative. The coherence in C-band QDs grown on the metamorphic buffer was also investigated in Ref. [23]. There, it was observed that, on average, T_2 increases from 73 ps to 176 ps when the excitation scheme is changed from off-resonant to resonant, indicating potential for future improvement of our structures.

While we cannot give a definite answer on the dephasing mechanism yet, spectral diffusion is more likely limiting the coherence in the present work. This is supported by the observation of random telegraphic noise during our experiments, i.e., sudden jumps in the emission energy on the timescale of seconds. This telegraphic noise was more pronounced at higher excitation powers - a higher number of excess charges (see Fig. R4), which further supports the dominant role of spectral diffusion. We could partly mitigate this effect by applying weak above-band light, although not consistently throughout our experimental study. In summary, this behavior suggests that spectral diffusion is likely to appear also at shorter timescales and/or with smaller magnitudes, thus representing the primary source of the reduced coherence time in our study. In the next generation of devices, we envision the implementation of a *p-i-n* junction to reduce the charge noise in the vicinity of the QDs and to control its charge state (see also answer to Comment 2, Reviewer #2).

To provide additional insight, we have also performed the HOM experiment at low excitation power for a 12.5 ns delay between subsequent photons (data not shown), yielding a reduced coherence time of 52 ps and reduced raw indistinguishability. This decrease in coherence time and photon indistinguishability with increasing temporal delay additionally points

towards non-Markovian noise processes affecting the emitter dephasing (Ref. [21]). A systematic study is required for a better understanding of the origin of noise processes, which is beyond the scope of this work.

Fig. R4. PL spectrum of QD-CBG device under quasi-resonant excitation as a function of excitation power. Each spectrum is recorded at increasing excitation power and integrated for 3 s; therefore, power and time are in direct correspondence. Thus, spectral jumps can be observed in time that become more likely at higher powers.

We have added the following explanation to the main text of the manuscript:

“From the width of the central dip we extract a photon coherence time of $T_2 = (103 \pm 13)$ ps, which, given the Purcell-reduced lifetime of $T_1 = 400$ ps, results in a T_2/T_1 -ratio of 0.44, comparing favorably with previous reports for QDs emitting in the telecom C-band^{22,23}. The still relatively short coherence time observed in our work is mainly attributed to fluctuating charges in the QD environment, suggested by the observed time-dependent spectral diffusion of emission lines, which most probably limits the observed indistinguishability. The coherence properties may be further improved by implementing electrical charge stabilization via electric gates or using droplet epitaxy as an alternative growth technique²².”

[21] A. Thoma, P. Schnauber, M. Gschrey, M. Seifried, J. Wolters, J. H. Schulze, A. Strittmatter, S. Rodt, A. Carmele, A. Knorr, T. Heindel, and S. Reitzenstein, “Exploring dephasing of a solid-state quantum emitter via time- and temperature-dependent Hong-Ou-Mandel experiments,” *Phys. Rev. Lett.* 116, 1–5 (2016).

[22] M. Anderson, T. Müller, J. Skiba-Szymanska, A. B. Krysa, J. Huwer, R. M. Stevenson, J. Heffernan, D. A. Ritchie, and A. J. Shields, “Coherence in single photon emission from droplet epitaxy and Stranski–Krastanov quantum dots in the telecom C-band,” *Appl. Phys. Lett.* 118, 014003 (2021).

[23] C. Nawrath, F. Olbrich, M. Paul, S. L. Portalupi, M. Jetter, and P. Michler, “Coherence and indistinguishability of highly pure single photons from non-resonantly and resonantly excited telecom C-band quantum dots,” *Appl. Phys. Lett.* 115, 023103 (2019).

5 - Regarding the lower purity and coherence times at higher powers, is the presence of other nearby QDs involved? While the devices feature single QDs emitting at a narrow filter window, there is still a relatively high density of QDs surrounding the positioned ones, with emission elsewhere, so it's not unreasonable to think that quasi-resonant excitation could be accessing such QDs.

The QD density in the structure used for imaging is $3.1 \times 10^8 \text{ cm}^{-2}$ (see Supplementary Note 2.), which can be translated onto an average of ~ 4 dots per central disk of the CBG with a radius of $R_0 = 648 \text{ nm}$. Only the central disk is illuminated by the excitation laser beam, with the expected beam waist below $2 \mu\text{m}$.

We consistently observe a single QD spectrum overlapping with the cavity mode under the quasi-resonant pumping condition at 0.835 eV (1484 nm) and low excitation power ($20 \mu\text{W}$). However, we cannot entirely reject the possibility that other QDs are also excited and emit in the spectral range outside the cavity mode.

The question is how the photo-induced states in other QDs can influence the coherence of a cavity-coupled QD. We believe that photo-induced neutral excitons or biexcitons in neighboring QDs have little effect on the coherence properties of an examined state. However, the appearance of charged excitons or biexcitons can be more effective. The charged states may be induced by non-geminate trapping of electrons or holes in the QD via photo-doping or due to background doping. While it seems unlikely that the former will occur under the quasi-resonant pumping condition, the latter is difficult to control and may significantly alter the coherence properties. To mitigate the effects of the fluctuating charge environment, we plan to introduce techniques such as further reducing the density of QDs or placing QDs in the *p-i-n* junction to control the QD charge state in the next generation of our devices.

The Reviewer raised a valid issue regarding the reduction in the purity of single-photon generation and coherence time at high pump power density. We believe that the problem is not directly linked to the excitation of many QDs within a central disk but rather stems from the photo-induced changes in the surrounding charge environment or the central disk sidewalls. To mitigate this problem, one can passivate the cavity sidewalls or employ the *p-i-n* junction to control the charge environment in the surrounding QD material.

(continued) I suppose in this case potentially growing lower densities of QDs could help - a related question, then, is whether lower QD densities would be a possibility.

As shown in Ref. [12], the density of QDs can be reduced to very low levels ($\sim 10^7 \text{ cm}^{-2}$) by employing the near-critical QD nucleation regime. This, however, comes at the cost of a reduced number of possible devices per imaging field.

Considering the QD density of $3.1 \times 10^8 \text{ cm}^{-2}$ and the central disc radius of $R_0 = 648 \text{ nm}$, the average number of QDs in a disc is ~ 4 , and in an imaging field, it is $\sim 3,300$. Therefore, only a tiny fraction of the QDs is optically active in the employed filtering range ($1550 \text{ nm} \pm 8 \text{ nm}$), as we register, on average, 10 QD emission spots per field. The median QD emission linewidth is 0.76 nm (see Supplementary Note 6, part A). The spectrally broad distribution of Stranski-Krastanov QDs (typical FWHM of about 180 nm) is a general challenge, while applied here near-critical (sub-critical) nucleation results in an even broader spectrum of

QDs due to the prolonged growth interruption applied. However, in our case, the broadening naturally helps to isolate the narrow emission lines from individual QDs spectrally.

In summary, the density used for the processing should constitute a trade-off between the sufficient isolation between QDs and the number of devices that can be fabricated per field.

We added the following sentence to the Discussion:

“The low QD density of $3.1 \times 10^8 \text{ cm}^{-2}$ guarantees their spatial isolation sufficient for the imaging procedure while maximizing the number of devices that can be fabricated per field.”

[12] Y. Berdnikov, P. Holewa, S. Kadkhodazadeh, J. M. Śmigiel, A. Frąckowiak, A. Sakanas, K. Yvind, M. Syperek, and E. Semenova, “Fine-tunable near-critical Stranski-Krastanov growth of InAs/InP quantum dots,” (2023), arXiv:2301.11008

6 - Relatedly, have the authors attempted LA-photon or direct resonant excitation of the QD? It would be interesting to see if the coherence time can be significantly improved, since these two excitation methods are minimally detrimental to the single-photon coherence times.

We attempted LA-phonon and strict resonant excitation, which would indeed give further insights into limiting factors. However, the necessary polarization filtering of the reflected laser light under resonant excitation is more challenging for micro- or nanophotonic structures such as CBGs than for planar samples due to the scattering of laser light. We did observe spectral signatures of resonance fluorescence. Unfortunately, HOM experiments have not been within reach under this excitation scheme due to the remaining amount of unsuppressed scattered laser light.

Noteworthy, we recently succeeded in two-photon resonant excitation on this type of QDs (Stranski-Krastanov) embedded in non-deterministically positioned mesa structures, and both the indistinguishability and coherence time of the generated photons increased significantly [34]. We added this information to the corrected manuscript:

~~*In combination with coherent pumping schemes, such as two-photon resonant excitation, we expect a further improvement of the photon coherence time and hence indistinguishability²⁶.*~~

“Implementing coherent optical pumping schemes, such as two-photon resonant excitation³⁶, also for scalably fabricated devices, is a crucial next step to further improve the photon coherence time and hence indistinguishability³⁷.”

Two-photon resonant excitation with deterministic QD-CBG devices has yet to be achieved.

[34] C. Nawrath, H. Vural, J. Fischer, R. Schaber, S. L. Portalupi, M. Jetter, and P. Michler, “Resonance fluorescence of single In(Ga)As quantum dots emitting in the telecom C-band,” *Appl. Phys. Lett.* 118, 244002 (2021).

[36] D. A. Vajner, P. Holewa, E. Zięba-Ostój, M. Wasiluk, M. von Helversen, A. Sakanas, A. Huck, K. Yvind, N. Gregersen, A. Musiał, M. Syperek, E. Semenova, and T. Heindel, “On-demand Generation of Indistinguishable Photons in the Telecom C-Band using Quantum Dot Devices,” (2023), arXiv:2306.08668

[37] A. Reigue, R. Hosten, and V. Voliotis, “Resonance fluorescence of a single semiconductor quantum dot: the impact of a fluctuating electrostatic environment,” *Semicond. Sci. Technol.* 34, 113001 (2019).

7 - What is limiting the post-selected visibilities for the different excitation powers, or why is there such large variability? Close to zero delay, well below the coherence time, it should be always very high, unless there’s a significant spatio-temporal and polarization mismatch between the two photons being interfered. I think polarization and spatial mismatch can be well controlled in the experiment, though polarization control can be tricky when using optical fibers. Is temporal mismatch, e.g. due to jitter cause by quasi-resonant excitation, an issue?

We are indeed confident that the spatio-temporal and polarization matching are well controlled in our HOM experiments. This we ensure by interfering laser pulses at the QD emission wavelength resulting in visibilities of about 97%, thus resulting in minor limitations of the post-selected visibility. The quasi-resonant excitation can introduce an additional timing jitter, which we cannot verify directly at this point.

Overall, the obtained post-selected TPI visibilities agree within the first or second error interval. As detailed below, the observed minor deviations can be explained by (1) the effect of the detector timing resolution in combination with a reduced coherence time, (2) the excitation-power dependence of $g^{(2)}(0)$, and (3) limited statistics:

Ad. 1 At saturation power, the extracted coherence time (80 ps) is already close to the timescale of our detector timing resolution (57 ps, cf. Fig. R5a). We refrained from including the system response in the HOM data analysis, as the presented raw values are most relevant for applications. Additionally, performing a deconvolution with the detector response increases the uncertainty when the count rates are small. However, accounting e.g., for our detector response by fitting the additional HOM data set acquired for 12.5 ns pulse-separation and at $0.25 \times P_{\text{Sat}}$ results in almost 100% post-selected visibility (dashed black line in Fig. R5b). From the convoluted fit, it is possible to separate the detector response from the TPI histogram, as shown in Fig. R5c.

Fig. R5. (a) Measured effective system response (dots) with Gaussian fit (blue line) of the superconducting nanowire single-photon detectors used in our HOM experiments, by correlating laser pulses of 5 ps duration. (b) Standard fit of the central dip to the co-polarized HOM data used in the manuscript (red line) and the deconvoluted fit including the detector response function (black dashed line). (c) Decomposition of the effective fit [panel (b)] into

the system response function (SRF) component (solid red line) and the deconvoluted HOM dip component (black dashed line).

Ad. 2 We did not correct the HOM visibility for limited purity for the same reasons for which we did not include the temporal response. The $g^{(2)}$ at zero time delay increases from 0.005 to 0.098 from low power ($0.02 \times P_{\text{Sat}}$) to saturation power ($1 \times P_{\text{Sat}}$), thus reducing the observed post-selected visibility at higher excitation powers. If we subtract the additional coincidences caused solely by the reduced purity from the post-selected visibility values following L. Zhai *et al.*, *Nat. Nanotechnol.* 17, 829–833 (2022), using the formula

$$V_{\text{TPI,corr.}} = [1 + 2g^{(2)}(0)] V_{\text{TPI, raw}}$$

we obtain purity-corrected post-selected TPI visibilities as summarized in Table R3.

Table R3. Purity-corrected post-selected TPI visibilities $V_{\text{TPI,corr.}}$

Excitation Power	$0.02 \times P_{\text{Sat}}$	$0.04 \times P_{\text{Sat}}$	$1 \times P_{\text{Sat}}$
$V_{\text{TPI,raw}}(0)$	0.80 ± 0.13	0.99 ± 0.06	0.84 ± 0.03
$g^{(2)}(0)$	0.006 ± 0.005	0.005 ± 0.003	0.098 ± 0.019
$V_{\text{TPI,corr.}}(0)$	0.81 ± 0.14	0.99 ± 0.07	1.00 ± 0.05

As one can see, correcting for the increase in multi-photon events at higher excitation powers (P_{Sat}) already fully accounts for the imperfect post-selected visibility. All values now agree within the first error interval. However, as this subtraction is not easily possible in real applications, we state the uncorrected values in the main text. While the extracted coherence time decreases when the excitation power is increased, the purity-corrected post-selected visibility is still ideal, $V_{\text{TPI,corr.}}(0) = 1.00 \pm 0.05$.

Ad. 3 Finally, the influence of statistics becomes apparent as the extracted post-selected visibility value at the lowest ($0.02 \times P_{\text{Sat}}$) excitation power has the largest error due to reduced statistics resulting from a low count rate, which is why the measurement at saturation power has the smallest uncertainty.

We have adapted the discussion of post-selected visibilities in Supplementary Note 7, part D, as follows:

~~“The extracted post-selected values are $V_{\text{PS}} = (80 \pm 13)\%$, $(99 \pm 6)\%$ and $(84 \pm 3)\%$ without clear dependence on the excitation power within the determined fit uncertainty and mainly limited by the detector timing resolution.”~~

“The variation between the extracted post-selected values of $V_{\text{PS}} = (80 \pm 13)\%$, $(99 \pm 6)\%$, and $(84 \pm 3)\%$ for different excitation powers results from the finite temporal resolution of our setup (57 ps FWHM jitter that is especially critical when resolving short coherence times), limited statistics due to reduced count rates at lower excitation powers, also evidenced by the larger error, and the increased multi-photon contributions at higher power. Correcting for the increase of $g^{(2)}(0)$ at higher powers, the extracted post-selected visibility values agree

within the standard errors. However, to be in line with future real-world applications, we state the uncorrected values.”

8 - The authors should further clarify what QDs were selected as references for the radiative rate enhancement estimates. E.g., were the 8 reference QDs located inside cavities, but spectrally detuned? Or were they completely outside any cavities? This is not very clear in the text, though it's somewhat important because there's a chance for radiative rate suppression of QDs located at resonance nodes in the cavity.

Thank you for pointing out this omission. We have clarified in the text that the QDs used as references are located outside of the cavities. Therefore, no significant modification of the radiative transition rate is expected, particularly no suppression due to the possible placement at the resonance node.

We have modified the manuscript text:

“To take statistical QD-to-QD fluctuations for the reference decay time into account, we estimate the average decay time of 8 ~~uncoupled~~ QD CX lines observed from dots located outside of the cavities, i.e. without Purcell-induced modification of the radiative lifetime, and obtain $\tau_{ref} = (1.99 \pm 0.16)$ ns, while the single reference shown in Fig. 3b has a decay time of (2.01 ± 0.02) ns.”

Reviewer #2 (Remarks to the Author):

In this experimental research paper, the authors presented the generation of single, indistinguishable photons emitted by localized InAs/InP QDs in the telecom C-band range. To date, a truly scalable technology for generating on-demand, high repetition rate single-photons at 1550 nm has remained elusive. This is despite the high relevance of the telecom C-band for quantum networks mediated by flying qubits.

In this report, the authors demonstrated deterministic coupling of pre-selected quantum emitters with circular Bragg gratings (CBG). These devices generate single photons with high purity and indistinguishability. The data presented were acquired using suitable methods, well analyzed, interpreted, and presented with good detail, ensuring their technical accuracy. The authors presented compelling evidence to support their assertions regarding high photon indistinguishability, reliable positioning, and Purcell effect. All necessary controls have been duly incorporated in the study.

We are very pleased with the Reviewer's comments on the relevance of our research direction at C-band for the development of quantum networks and pointing out that some technical achievements presented outside the telecom range have yet to be shown. Further, we are thankful for the positive evaluation concerning the overall quality, which, according to the Reviewer, is sufficient to support the main points of the manuscript.

However, as elaborated below, the results are mainly incremental with low potential to advance the field. As such, publication in Nature Communications is NOT recommended.

We politely disagree with the Reviewer's judgment. Telecom C-band quantum photonic device deterministic and scalable fabrication with 30% yield and observation of HOM interference are substantial advancements in the field, motivating and inspiring further efforts in quantum photonics research at the telecom wavelength.

1) The demonstrated positioning involves CBG, which possesses a quite large mode volume (V) and a low-quality factor (Q) (fig 3a). While this is useful for showcasing deterministic coupling, the most crucial quantum devices necessitate an ultra-high Q/V ratio. This is evident also in the low Purcell factor reported.

We cannot agree with the Reviewer's statement that "*the most crucial quantum devices necessitate an ultra-high Q/V ratio,*" and there is no evidence of this statement in the literature. The required device parameters in terms of the Q/V ratio and the Purcell factor strongly depend on the intended application. There are several crucial parameters for a cavity-coupled single-photon source, and some trade-offs between them are necessary to fulfill the device specification for a particular purpose. So, balancing between these parameters is essential for enabling a superior quality device with a specific functionality.

For example, there is a well-known trade-off between photon extraction efficiency and the quality factor Q mentioned by the Reviewer. Increasing the Q factor beyond the critical coupling regime (at constant mode volume) reduces the photon extraction efficiency due to effective light confinement in the cavity and the elevated impact of the intracavity losses. Consequently, high Q factor cavities reduce the source brightness. However, brightness is crucial for any application utilizing the photons as a resource, including schemes for quantum communication and (distributed) quantum computation utilizing optical networks and flying qubits.

It is known that high-Q and low-V cavities can achieve the strong coupling condition. Once reached, the source can operate in a photon blockade mode, effectively suppressing multi-photon events, thus providing photon generation with high purity [K. Müller *et al.*, *Phys. Rev. Lett.* 114, 233601 (2001)]. In addition, the strong coupling condition can considerably speed up the emission, switching off most phonon-mediated decoherence channels, leading to high photon indistinguishability—the source's high speed and coherence suit quantum communication and quantum computation applications. However, a strong interaction of an emitter with the cavity mode opens new decoherence mechanisms related to the excitation of higher states in the Jaynes-Cummings ladder, deteriorating the source coherence. Again, there is a trade-off between the coupling regime and the required coherence level of the source. Because of that, implementing the weak coupling regime can be sufficient.

However, our decision to use CBGs in this work has several important reasons, as detailed in the following part of our answer to this Question.

Advantages of using CBGs

For some purposes, especially for exploring fundamental physics, it is indeed important to maximize the Q/V ratio for reaching the strong coupling regime and/or exploiting strong non-linearities/phase shifts. On the contrary, applications that utilize photons as a resource, a high photon-extraction efficiency, and a robust spectral response (in terms of bandwidth) are most important. For such applications, CBGs are, in fact, one of the most suitable and useful structures available to date. Contrary to the Reviewer's opinion, we believe that a CBG is an interesting photonic structure that features high levels of photon extraction efficiency in a broad spectral range (Fig. 1b and Ref. [14]). Moreover, the CBG is effectively an antenna structure with moderate Q factors (110 in our case) that results in Purcell enhancement with a spectrally broad response (Ref. [14]; here, FWHM of 14 nm). Therefore, the spectral mismatch problem between the emitter and cavity resonance is greatly eliminated compared to micropillars or other high-Q cavities. This is important in general for single-photon sources, as well as for the generation of polarization-entangled photon pairs. In the widely-employed case of utilizing the biexciton-exciton radiative cascade to generate polarization-entangled photon pairs [see Akopian *et al.*, *Phys. Rev. Lett.* 96, 130501 (2006), Huber *et al.* *J. Opt.* 20 073002 (2018)], increasing the radiative transition rate for both lines and bringing their ratio close to 1 is indispensable. This can be achieved using the CBG approach (the biexciton binding energy in the employed QDs of 3.2 meV corresponds to ~7 nm of the separation of the lines), which would be impossible to achieve using a high-Q micropillar.

Suitability of CBGs in quantum communication protocols

Our design can theoretically provide a Purcell factor as high as $F = 18$, and β factor of $(F-1)/F = 94.4\%$, allowing to harvest almost 95 % of the emission with the cavity, and to shorten the radiative decay to levels comparable with the exciton coherence time, which is indispensable for many applications in quantum information processing, as long as reaching the strong light-matter coupling regime is not needed. For example, our work aims to explore the quantum optical properties of deterministically fabricated telecom QD-CBGs for applications in quantum communication protocols, which do not require extreme Purcell enhancements. CBGs can achieve Purcell factor > 20 in simulations even when fiber-coupled (Ref. [15]), while factors > 10 have already been demonstrated experimentally [Wang *et al.*, *Phys. Rev. Lett.* 122, 11 (2019)].

CBG design provides the best solution for out-of-plane emission geometry also from the perspective of mechanical stability, robustness, small size, and easy fabrication procedure. We have developed a fabrication technology based on wafer bonding and implemented a bottom metallic mirror, as reported in Ref. [10], to increase emission directionality. This is beneficial for both the localization of QDs due to the spectrally broad enhancement of the photon extraction (Ref. [10]) and for the final stage of the device fabrication, as it is an integral part forming a hybrid CBG.

[10] P. Holewa, A. Sakanas, U. M. Gür, P. Mrowiński, A. Huck, B.-Y. Wang, A. Musiał, K. Yvind, N. Gregersen, M. Syperek, and E. Semenova, "Bright Quantum Dot Single-Photon Emitters at Telecom Bands Heterogeneously Integrated on Si," *ACS Photonics* 9, 2273–2279 (2022).

[14] L. Bremer, C. Jimenez, S. Thiele, K. Weber, T. Huber, S. Rodt, A. Herkommer, S. Burger, S. Höfling, H. Giessen, and S. Reitzenstein, "Numerical optimization of single-mode fiber-coupled single-photon sources based on semiconductor quantum dots," *Opt. Express* 30, 15913 (2022).

[15] L. Rickert, F. Betz, M. Plock, S. Burger, and T. Heindel, "High-performance designs for fiber-pigtailed quantum-light sources based on quantum dots in electrically-controlled circular Bragg gratings," *Opt. Express* 31, 14750–14770 (2023).

2) This work primarily builds upon the existing literature and previous publications from the same group. (i) The positioning of QDs to photonic resonators (such as CBG) has been demonstrated multiple times using the same PL technique. (ii) Besides the SPS (already showed by several groups and in ref. 10 by the authors), this paper has not demonstrated the versatility of this structure in enabling new quantum devices. For example, it is unclear if the CBG can be tuned independently from the QDs or vice versa. Additionally, questions remain regarding the possible implementation of quantum optics schemes, such as a quantum repeater, or the ability to voltage-control the QD exciton or trigger emission. In this regard it is not a breakthrough that significantly extends the knowledge and fosters innovation, ultimately contributing to the development of new technologies, quantum protocols, or device research.

We appreciate the Reviewer's opinion on implementing advanced control and application schemes. This gives us the chance to further stress the high degree of novelty of our work. In fact, in this work, we succeeded in combining several concepts, previously demonstrated individually, in a single device, enabling an unprecedented performance of quantum light sources in the application-relevant telecom C-band while also presenting a route that solves the random nucleation problem of otherwise state-of-the-art self-assembled QDs. In detail, these concepts are:

1. Fabrication of the sample for increased SNR at C-band, using the metallic mirror (as the first group in the world),
2. Performing the localization of QDs operating in the C-band (as the first group in the world),
3. Fabrication of CBGs working in the C-band (as the second group in the world and the first group fabricating CBG in the C-band deterministically),
4. Maximizing the photon indistinguishability for QDs at C-band wavelengths (we hold the record).

While the positioning of QDs has indeed been demonstrated for other spectral ranges, e.g., Ref. [25], we succeeded in adapting the scalable technologies to wavelengths around 1550 nm, which is a challenge on its own (see Question 1, Reviewer #1). For this, we specifically had to design the structure to enable imaging with $\text{SNR} > 1$, requiring a specific sample geometry and which is very different from the concept presented in Ref. [25]. This achievement represents a qualitative change and is of highest relevance for the photonics community requiring active quantum photonic devices operating in the telecom C-band for the development of practical schemes for communication and computation. As outlined in our manuscript, the technical difficulties that have so far been associated with shifting the technology to telecom (2D detector noise level, QD structure suitable for fabrication, performance of optical elements operating at telecom), as well as the design of CBGs

operating at telecom, have been successfully addressed here, contributing a substantial advancement to the field.

QD and CBG tunability

Moreover, we can convincingly address the important point of the spectral tunability of the CBG mode relative to the QD emission mentioned by the Reviewer. We investigated this question during fabrication and established that scaling the central disc diameter together with shifting the grating results in a pronounced cavity mode energy shift (see Fig. R6). As the dependence is linear in approximately 150 nm-wide tuning range, this provides a straightforward way to target a specific QD emission energy. This enables imaging of the PL emission using versatile narrow bandpass filters and fabricating the dedicated (scaled) CBG in the same field, further increasing the number of devices per field without changing the QD density.

Fig. R6. Tuneability of the CBG cavity mode energy. (a) Stacked PL spectra for cavities with the radius R changing in steps of 2%. Inset: The distance between the grating and the mesa is constant while scaling the R . (b) Extracted cavity mode energy for spectra shown in (a). R_0 - the optimized central mesa diameter.

In the next generation of devices, we aim to implement electrical control of the QD emission. This requires a $p-i-n$ structure such as, for example, applied in the InAs/InP QD system to electrically inject the carriers [T. Müller *et al.*, *Nat. Commun.* 9, 862 (2018), Z.-H. Xiang, *Commun. Phys.* 3, 121 (2020)]. In our case, we expect the $p-i-n$ structure to reduce charge noise in the QD vicinity (Refs. [7] and [35]), thus increasing the photon coherence time and enabling electrical field tuning via the Stark effect. We will also investigate reducing the exciton fine structure splitting in analogy to H. Ollivier *et al.*, *Phys. Rev. Lett.* 129, 057401 (2022). The required electrical contacts can also be made on the CBG to the central pillar with minimal impact on the optical properties [Ref. [13] and Q. Buchinger *et al.*, *Appl. Phys. Lett.* 122, 111110 (2023)]. Alternatively, one could tweak the exciton emission energy using the strain tuning of the QD (T. Lettner *et al.*, *Nano Lett.* 21, 24 (2021).

In summary, we can reliably fabricate the antennas operating at 1550 nm but acknowledge that the subsequent step is the implementation of the QD tuning schemes.

We have included this information in the corrected manuscript:

“It is thus important to stabilize the QD environment, e.g. via electrical gates, which also enable the tunability of the QD emission energy.”

Quantum repeaters and other advanced schemes

Concerning the research on quantum repeaters using deterministic QD sources mentioned by the Reviewer - this field has so far focused almost entirely on wavelengths outside the telecom range. A quantum repeater is highly ambitious and will require many photons with pairwise entanglement, high purity, and high indistinguishability. After improving on these parameters, we certainly will consider implementing repeater schemes.

In summary, we believe that deterministic fabrication, as reported in our manuscript, is the key enabling element for practical quantum photonic devices operating in the highly relevant telecom C-band, irrespective of their specific application. We are well aware of current limitations, and as outlined above, we have a strategy to address these.

[7] N. Tomm, A. Javadi, N. O. Antoniadis, D. Najer, M. C. Löbl, A. R. Korsch, R. Schott, S. R. Valentin, A. D. Wieck, A. Ludwig, and R. J. Warburton, “A bright and fast source of coherent single photons,” *Nat. Nanotechnol.* 16, 399–403 (2021).

[13] A. Barbiero, J. Huwer, J. Skiba-Szymanska, T. Müller, R. M. Stevenson, and A. J. Shields, “Design study for an efficient semiconductor quantum light source operating in the telecom C-band based on an electrically-driven circular Bragg grating,” *Opt. Express* 30, 10919 (2022).

[25] L. Sapienza, M. Davanço, A. Badolato, and K. Srinivasan, “Nanoscale optical positioning of single quantum dots for bright and pure single-photon emission,” *Nat. Commun.* 6, 7833 (2015).

[35] A. V. Kuhlmann, J. Houel, A. Ludwig, L. Greuter, D. Reuter, A. D. Wieck, M. Poggio, and R. J. Warburton, “Charge noise and spin noise in a semiconductor quantum device,” *Nat. Phys.* 9, 570–575 (2013).

3) The authors gave a certain emphasis (title) to the scalability of their approach. Scalability requires a clear context for better understanding and accurate interpretation. Without such knowledge, determining the scalability of the structure remains ambiguous.

Here, it is unclear if the structure needed to optimize the localization (Fig. 1a and Fig. 2b) is tolerant with the implementation of real quantum devices in terms of scalable integration and manufacturing. Scalability in terms of integrating other functional photonic devices would rely on the compatibility of structures and materials. In this regard, the authors should show how that QD-CBG structure in presence of metal markers can be coupled to a heterogeneous nanophotonic structure. Thorough analysis and testing should be conducted to ensure seamless interaction and optimized performance.

We thank the Reviewer for pointing out the different meanings and differences in understanding of the word *scalability*. The interpretation presented by the Reviewer focuses on integrating our devices with another platform. In other words, there is an assumption that we are missing an additional step in processing our devices. Specifically,

the Reviewer writes (...) *the authors should show how that QD-CBG structure in presence of metal markers can be coupled to a heterogeneous nanophotonic structure.*

However, the heterogeneous integration of the QD-CBG structure has never been our aim. As our devices are optimized for vertical light emission, the integration-relevant next point should rather be attaching an optical fiber to the cavity instead of the multi-layer integration as known from, for example, the III-V/Si heterogeneous systems. If we fabricated such devices, we would need to test the repeatability of the integration process, for example, micro-transfer printing. Still, this understanding of the term *scalability* focuses on another stage of processing the QD samples, which has never been a central concern in the community.

What has hindered the progress in the field so far is the scalability of embedding a QD in a photonic device, not the scalability of integrating the fabricated nanophotonic device with a QD with the target platform. The reason was the need for a qualitative change in the approach to the device fabrication that was missing at 1.55 μm until now, as the localization of QDs is relatively specific for this class of randomly placed point emitters. In contrast, the scalability of the device integration could be adapted from other fields that transfer the active material into the Si chip.

Therefore, our understanding of the term scalability refers to the QD localization process and is justified by comparison with other localization methods, such as scanning cathodo- or photoluminescence, which are much slower. Our interpretation is supported by Reviewer #3: *The central point of the contribution relates to scalability. Namely, many different photonic structures can be fabricated to enable higher collection efficiency or Purcell factor.*

(continued) Scalability in terms of manufacturing refers to the ability to produce large volumes of these photonic devices while maintaining consistent quality. However, self-assembled QDs nucleate on the surface in random location and with different exciton emission spread on the spectrum. Thus, the QD-CBG structure will emit photon of different wavelengths with different exciton configuration and properties. The presented approach must go through a non-parallelizable sorting of the QDs.

We are thankful for touching on these challenges inherent to the self-assembled QDs. Our approach is to accept the disadvantages of working with these emitters, as the advantages outnumber them, and to present the workaround for the challenges so that they do not prevent the successful operation of the device.

The Reviewer writes (...) *self-assembled QDs nucleate on the surface in random location and with different exciton emission spread on the spectrum.* We agree that these are two significant challenges to overcome once one intends to employ self-assembled QDs in any device using a reliable process flow. Our work demonstrates how to eliminate both problems: we address the random nucleation by localizing the QDs and select the QD emission wavelength using a narrow band-pass filter while recording the μPL maps to handle the variations in exciton emission energy. The spectral distribution of our devices can be minimized with an even narrower band-pass filter. In this sense, prior to the cavity fabrication and during the imaging process, we can select only these QDs that match very narrow emission windows and thus are spectrally standardized, opening the way for the fabrication

of large volumes of photonic devices while maintaining consistent quality - as the Reviewer claimed.

Regarding the *different exciton configurations*, it is mostly identical across the chip - in our case, we have a weak unintentional doping of the InP matrix, resulting in trions dominating the emission spectrum of each device. Additionally, our ongoing efforts to embed the QDs in a *p-i-n* junction will provide control over the exciton configuration so that the QD is depleted of resident carriers and exciton and biexciton dominate the μ PL spectrum if needed.

The Reviewer also writes *The presented approach must go through a non-parallelizable sorting of the QDs*. Still, it is unclear which method would be superior for the localization of the emitters. To find them, one must search on the part of the chip with QDs, which is, in principle, inherently non-parallelizable. Site-selective growth of QDs could be utilized instead, where the patterning of the wafer determines the QD nucleation. However, this is an entirely different approach, and such QDs lack demonstration of superior quantum optical properties compared to QDs obtained via the self-assembled growth approach. Moreover, site-selective growth provides only spatial ordering, not eliminating the spectral dot-to-dot variation due to the fluctuation of material properties and QD dimensions at each QD position (reflected in their emission energy). Consequently, the μ PL imaging process/spectral pre-selection is still required before the complete device fabrication.

Therefore, if we focus on working with self-assembled QDs, the localization can be done using slow scanning techniques or fast imaging of the large chip area. Since recording a μ PL map takes only about 2 seconds, with an automatized and programmed cryostat stage, the entire localization procedure can be unsupervised - an operator involvement would be reduced to mounting and cooling down the sample and setting the imaging parameters. As a result of the procedure, one would quickly get the μ PL maps with fitted QD positions, which is essential in the context of the manufacturing approach as defined by the Reviewer (*ability to produce large volumes of these photonic devices while maintaining consistent quality*).

For example, one could apply our method on a standard 2" InP wafer intended to be divided into ~65 chips of (5x5) mm². The centrally located 25 fields of (50x50) μ m² on each chip can be imaged, while only one optimal QD spot per chip can be chosen for cavity definition. Even if we include the stage movement and assume 4 s is needed per field, imaging the discussed 1625 fields will take less than 2 h. After device fabrication, cleaving into chips, and fiber coupling, we expect to have as many fiber-coupled sources of indistinguishable single photons as fabricated chips. A similar concept with a localization approach was applied in Ref. [28] and has become the foundation of the Quandela company which, i.a., sells fiber-coupled single-photon sources operating at ~ 900 nm.

[28] A. Dousse, L. Lanco, J. Suffczyński, E. Semenova, A. Miard, A. Lemaître, I. Sagnes, C. Roblin, J. Bloch, and P. Senellart, "Controlled light-matter coupling for a single quantum dot embedded in a pillar microcavity using far-field optical lithography," *Phys. Rev. Lett.* 101, 30–33 (2008).

Reviewer #3 (Remarks to the Author):

In the manuscript entitled “Scalable quantum photonic devices emitting indistinguishable photons in the telecom C-band” by Holewa et al. the authors report on a quantum dot-based source of single photons. The source can emit photons at ~1550nm and the collection efficiency is enhanced using Circular Bragg Grating (CBG) cavity.

The central point of the contribution relates to scalability. Namely, many different photonic structures can be fabricated to enable higher collection efficiency or Purcell factor. However, depending on the complexity of the fabrication the yield of functioning structures is commonly quite low. Furthermore, most of the structures need what is commonly known as deterministic fabrication – the photonic structure needs to be accurately placed over the site of formation of the emitter, otherwise the collection efficiency and Purcell effect will not be achieved.

The CBG cavities are photonic structures known to be notoriously hard to fabricate deterministically, which in return leads to reduced value of the Purcell enhancement compared to the design value. The most complicated element of the CBG implementation is elimination of birefringence that is known to reduce the applicability of the device – one cannot generate entangled photon pairs nor get a very good value of the two-photon interference.

While this was shown for the devices operating in wavelength range 750-950, a deterministic CBG device was never shown in telecom range. One of the reasons for this is that the emitter imaging in the wavelength range is very problematic. The imaging cameras are conceptually different – the pixel size is larger, and noise is stronger, which makes the imaging quite demanding. However, the authors have overcome this problem and reached the placement accuracy of ~90nm.

We would like to thank the Reviewer for the overall positive assessment of our work and for stressing its relevance to the photonic community.

Having summarized the presented results; I must conclude that submitted work might belong to portfolio of Nature Communications however, some issues need to be correctly addressed:

1. Could you please estimate the improvement needed in imaging to reach the accuracy where the collection efficiency and Purcell enhancement start to be more like what is predicted in the device theoretical simulation?

The response to this question is different concerning the Purcell enhancement and the photon extraction efficiency.

Regarding the photon extraction efficiency, the reason for the lower measured value compared to the simulated one is the same as for Comment 2, Reviewer #1, and does not depend on the dipole displacement ρ , at least for the simulated range of $\rho < 600$ nm. We have introduced the explanation of the discrepancy between calculated and measured

photon extraction efficiency to Supplementary Note 5., as indicated in response to Comment 2, Reviewer #1.

Regarding the Purcell enhancement, the simulated Purcell factor F decreases as soon as $\rho \neq 0$, reaching ~ 15 at $\rho = 50$ nm and half of its maximal value (~ 9) at $\rho = 100$ nm. However, the QDs chosen as exemplary (QD-CBG devices #1-#3) had a localization accuracy of 140 nm, and this dipole displacement corresponds to $F \sim 5$.

Therefore,

1. We could observe QDs with a higher Purcell factor if we had a higher process yield. It could be increased, e.g., by using a narrower bandpass filter during the imaging. Doing so would lower the number of detected QD spots per field. However, they would better match the central mode energy of the cavity.
2. For a better QD material quality resulting in higher brightness of QD spots, we potentially could increase the localization accuracy to 50 nm (as was shown for short wavelength QDs, Refs. [24] and [25]), and by doing so, reach values of ~ 15 for the Purcell enhancement.
3. As the Purcell factor dependence on the dipole displacement is considerable in the discussed range, the cavity design could also be changed so that the QD couples to the fundamental mode of the cavity. If the cavity mode had no in-plane nodes, the tolerance of the Purcell factor for the QD displacement would be higher.

[24] T. Kojima, K. Kojima, T. Asano, and S. Noda, "Accurate alignment of a photonic crystal nanocavity with an embedded quantum dot based on optical microscopic photoluminescence imaging," *Appl. Phys. Lett.* 102, 011110 (2013).

[25] L. Sapienza, M. Davanço, A. Badolato, and K. Srinivasan, "Nanoscale optical positioning of single quantum dots for bright and pure single-photon emission," *Nat. Commun.* 6, 7833 (2015).

2. Can you provide the lifetime fit made using linear scale. The background in the reference dot signal looks high enough to alter the log-fit result and artificially extend the lifetime. This could lead to an erroneous estimate of the Purcell enhancement.

Fitting of all decays reported in the manuscript and in the Supplementary Material was performed without calculating the logarithm of the PL signal - we used the linear scale instead. The used function was $I = A \cdot \exp(-(t-t_0)/\tau)$, where I is the signal intensity, A is the scaling factor, t is time, and τ is the decay time. In this way, the high-intensity part of the decay curve has a much higher weight for the fit than the low-intensity part, which merges with the background. We have plotted it in Fig. 3b using the logarithmic scale to better present the signal intensity change over two orders of magnitude and underline the monoexponential character of the decay.

In Fig. R7(a), we present the same TRPL traces as shown in Fig. 3b in the manuscript, plotted using a linear scale and stacked for clarity. Additionally, we show the regular residuals of the fits in Fig. R7(b) the regular residuals of the fits. However, we decided to keep the log scale in the manuscript as it is more helpful to showcase the agreement between the fit and the data points.

Fig R7. Linear fitting of the TRPL data. (a) Data points and fits as in Fig. 3b but in linear scale, (b) Fit residuals plot.

3. The two-photon interference is not very high, and this is attributed to the magnetic and electrical field fluctuations. What about the birefringence? Can the authors comment on the birefringence induced by the structure?

We agree with the Reviewer that the two-photon interference visibility still leaves much room for improvement, especially compared to the short-wavelength counterparts. However, we would like to stress that our work reports a substantial increase compared to previous work using QDs on metamorphic buffer layers (in the GaAs system) and achieving visibilities of up to 14% even under resonant QD excitation (Ref. [34]).

Birefringence is a known issue influencing the polarization state of the photons emitted from, e.g., nanowire QDs [Versteegh *et al.*, *Nat. Commun.* 5, 5298 (2014), K. D. Jöns, *et al.*, *Sci. Rep.* 7, 1700 (2017)], which can be accounted for to distill the polarization-entangled photon pairs. In the case of CBGs, the polarization state of the emitted photons can be affected by the radial spatial position of the quantum emitter relative to the center of the central disk. This effect is currently being investigated in the community [G. Peniakov *et al.*, arXiv:2308.06231 (2023)]. It arises from the coupling to differently polarized optical modes rather than mere birefringence.

The main impact of birefringence in the SPS application is to introduce spectral misalignment for one of the two dipoles of the trion emission - Supplementary Eq. 5, thus reducing the extraction efficiency without influencing the photon indistinguishability.

The CBG structure is susceptible to fabrication imperfections, and even 1 % ellipticity of the central mesa results in birefringence with a cavity mode splitting of ~ 10 nm [H. Wang *et al.*, *Nat. Photonics* 13, 770–775 (2019)]. Regarding the birefringence of our devices, the splitting of the orthogonal cavity modes is typically below 200 μeV . For device #2, it reaches 64 μeV , which is only 1.6% of the typical cavity mode width of 8 meV (as shown in Fig. 2a), evidencing a minute deviation from a cylindrically symmetric cavity.

REVIEWER COMMENTS

Reviewer #1 (Remarks to the Author):

I find that the authors' responses to all my comments are satisfying, though I have a couple of additional small comments below that should be addressed. I think all the changes the authors made to the main text and SI are good.

Regarding the response to my Comment 2: The authors claim that non-radiative paths created by structural defects created during growth, or induced by etching. I think the authors should expand on how these factors are not show-stoppers for this integration platform. It sounds a bit like these are things that cannot be avoided - is that the case, or there are reasonable, rational paths forward toward improving defects?

I agree with referee #2 that the word "scalability" in the title can be somewhat misleading, or at least there is too much nuance involved that it probably does not belong there. I would suggest maybe replacing the word with "high-throughput", which is essentially what the positioning technique provides in comparison with other, current techniques used for the same purpose. High throughput is necessary but not sufficient for device fabrication scalability, and the term is less nuanced. It's worthwhile pointing out that control of a single emitter spatial location is a tall hurdle towards scalability for any solid state quantum emitter material system, and is yet to be overcome. To some extent, heterogeneous optical properties is a less severe problem for e.g., color centers, however still also a barrier towards device scalability.

After the authors address these small comments, my opinion is that the manuscript is acceptable for Nature Communications.

Reviewer #2 (Remarks to the Author):

1) However, as elaborated below, the results are mainly incremental with low potential to advance the field. As such, publication in Nature Communications is NOT recommended.

--

We politely disagree with the Reviewer's judgment. Telecom C-band quantum photonic device deterministic and scalable fabrication with 30% yield and observation of HOM interference are substantial advancements in the field, motivating and inspiring further efforts in quantum photonics research at the telecom wavelength.

QDs emitting around 1550 nm have been a topic of research for at least the last 20 years. There have already been many reports in literature on the realization of efficient single photon emission with high purity using InAs/InP or InAs/GaAs structures. In InAs/InP QDs, values of $g(2)(0) \sim 4 \times 10^{-4}$ have been reported [e.g., T. Miyazawa et al., Appl. Phys. Lett. 109(13), 132106 (2016)], and secure key transmission over long distances (> 100 km) through optical fiber has been achieved. Single photon sources are characterized by $g(2)(0) \sim 0$ and photon indistinguishability. HOM measurement has been performed using either CW or pulsed excitation [e.g., C. Santori, D. Fattal, and Y. Yamamoto, "Indistinguishable photons from a single-photon device," Nature 419, 594 (2002)].

Achieving a 30% yield is commendable, but it is more of an incremental advancement rather than an innovative breakthrough.

3) The authors gave a certain emphasis (title) to the scalability of their approach. Scalability requires a clear context for better understanding and accurate interpretation. Without such knowledge, determining the scalability of the structure remains ambiguous.

Here, it is unclear if the structure needed to optimize the localization (Fig. 1a and Fig. 2b) is tolerant with the implementation of real quantum devices in terms of scalable integration and manufacturing. Scalability in terms of integrating other functional photonic devices would rely on the compatibility of structures and materials. In this regard, the authors should show how that QD-CBG structure in presence of metal markers can be coupled to a heterogeneous nanophotonic structure. Thorough analysis and testing should be conducted to ensure seamless interaction and optimized performance.

--

The heterogeneous integration of the QD-CBG structure has never been our aim.

I am glad the authors acknowledge their structure's integration limitations, which is a critical barrier to its potential. Nevertheless, in their reply, they propose integrating the QD-cavity into a p-i-n diode to enable carrier injection, indicating an understanding of the necessity for integration in enhancing device functionality.

Our understanding of the term scalability refers to the QD localization process.

Localization of QDs and their coupling with cavities was reported almost 20 years ago, but the QD-cavity system has not been considered 'scalable' by the community working in this field.

This is because the QDs nucleate randomly on the surface and the large inhomogeneous broadening inherent in the self-assembly process. This means that:

- i) each QD must be located separately,
- ii) each cavity must be designed with a different resonance.

Only a structure with spatially ordered QDs and small inhomogeneous broadening can be considered a scalable system in the sense meant by the authors. For instance, site-controlled InGaAs pyramidal QDs exhibit such characteristics, even though they come with other issues.

What has hindered the progress in the field so far is the scalability of embedding a QD in a photonic device, not the scalability of integrating the fabricated nanophotonic device with a QD with the target platform.

Again, the localization of QDs and their coupling with cavities was reported almost 20 years ago. What has prevented the field from moving forward is the large inhomogeneous broadening inherent in self-assembled QDs, that is, QDs emit single photons at different wavelengths.

For the specific device the author has in mind, a standalone single-photon source, a key requirement is low uncertainty in the emission wavelength. For example, to scale photonic

systems that require parallel photon generation, these sources must emit at uniform wavelengths, or the inhomogeneous broadening should be significantly less than the intrinsic linewidth.

6 - Relatedly, have the authors attempted LA-photon or direct resonant excitation of the QD? It would be interesting to see if the coherence time can be significantly improved, since these two excitation methods are minimally detrimental to the single-photon coherence times.

--

We attempted LA-phonon and strict resonant excitation, which would indeed give further insights into limiting factors. However, the necessary polarization filtering of the reflected laser light under resonant excitation is more challenging for micro- or nanophotonic structures such as CBGs than for planar samples due to the scattering of laser light. We did observe spectral signatures of resonance fluorescence. Unfortunately, HOM experiments have not been within reach under this excitation scheme due to the remaining amount of unsuppressed scattered laser light.

Resonant scattering in single QDs and in single QDs coupled to microcavities has been documented in many papers now. It has been reported even in structures (such as 2D L3 photonic crystals) that present a mode-mismatch with the incoming gaussian laser beam much larger than in CBGs.

The demonstrated positioning involves CBG, which possesses a quite large mode volume (V) and a low-quality factor (Q) (fig 3a). While this is useful for showcasing deterministic coupling, the most crucial quantum devices necessitate an ultra-high Q/V ratio. This is evident also in the low Purcell factor reported.

--

There are several crucial parameters for a cavity-coupled single-photon source, and some

trade-offs between them are necessary to fulfill the device specification for a particular purpose.

The CBG has a large mode volume, and the reported Purcell factor (F_p) is clearly quite low. F_p is proportional to Q/V . Because the authors focus on devices based on the weak coupling regime (such as single photon sources), they should still show a sizable Q/V . High efficiency is very important, but several emitter-cavity structures have already been shown to maintain high Q/V while achieving high vertical extraction (as seen in micropillar configurations) or lateral extraction (as integrated with 2D photonic crystals).

It is known that high- Q and low- V cavities can achieve the strong coupling condition. Once reached, the source can operate in a photon blockade mode, effectively suppressing multi-photon events, thus providing photon generation with high purity [K. Müller et al., Phys. Rev. Lett. 114, 233601 (2001)]. In addition, the strong coupling condition can considerably speed up the emission, switching off most phonon-mediated decoherence channels, leading to high photon indistinguishability—the source's high speed and coherence suit quantum communication and quantum computation applications. However, a strong interaction of an emitter with the cavity mode opens new decoherence mechanisms related to the excitation of higher states in the Jaynes-Cummings ladder, deteriorating the source coherence. Again, there is a trade-off between the coupling regime and the required coherence level of the source. Because of that, implementing the weak coupling regime can be sufficient.

This paper aims to demonstrate efficient single-photon emission. I surely agree that it is not beneficial for an emitter-cavity system meant for single-photon generation to be far into the strong coupling regime, and I did not mean that. Increasing Q/V will not necessarily bring one into the strong coupling regime.

Consequently, high Q factor cavities reduce the source brightness.

To a certain extent, because the brightness will be also improved by the enhanced spontaneous emission, which means high F_p .

However, a crucial figure of merit is the overall out-coupling efficiency, which measures how effectively internally generated single photons are transmitted into a desired single mode.

This efficiency approaches one when F_p is large and there is minimal loss of cavity photons to modes other than the desired one.

Contrary to the Reviewer's opinion, we believe that a CBG is an interesting photonic structure that features high levels of photon extraction efficiency in a broad spectral range (Fig. 1b and Ref. [14]). Moreover, the CBG is effectively an antenna structure with moderate Q factors (110 in our case) that results in Purcell enhancement with a spectrally broad response (Ref. [14]; here, FWHM of 14 nm).

--

CBG is an interesting structure that has been studied for many years. However, I do not believe that the findings presented in this paper, considering what is already in the literature (including publications by the authors), are substantial enough to warrant publication in Nature Communications.

This work primarily builds upon the existing literature and previous publications from the same group. (i) The positioning of QDs to photonic resonators (such as CBG) has been demonstrated multiple times using the same PL technique. (ii) Besides the SPS (already showed by several groups and in ref. 10 by the authors), this paper has not demonstrated the versatility of this structure in enabling new quantum devices. For example, it is unclear if the CBG can be tuned independently from the QDs or vice versa. Additionally, questions remain regarding the possible implementation of quantum optics schemes ... or the ability to voltage-control the QD exciton or trigger emission. In this regard it is not a breakthrough that significantly extends the knowledge and fosters innovation, ultimately contributing to the development of new technologies, quantum protocols, or device research.

--

The authors' reply did not adequately address my criticisms above. It is crucial to achieve fine, independent tuning, whereas Figure R6 indicates that the device exhibits only an irreversible, coarse adjustment. Several research groups have demonstrated reversible, independent tuning of the emitter and cavity using techniques such as the Stark effect (for the emitter) and layer deposition (for adjusting the cavity). The same authors acknowledge the necessity to implement an electrical tuning combined with carrier injection: "In the next generation of devices, we aim to implement electrical control of the QD emission. This requires a p-i-n structure such as, for example, applied in the InAs/InP QD system to electrically inject the carriers."

I encourage the authors to further develop their proposed device to clearly demonstrate its versatility and potential advantages. However, I have concerns regarding the integration of a p-i-n structure within the current design. I suspect that this addition may further reduce the quality factor (Q) of the device due to the absorption by free charges.

To critically enhance the significance of this paper and align it with the standards expected by Nature Communications, the authors should consider the following recommendations:

- 1) **Comprehensive Characterization of Emitters**: A detailed study of the emitters before integrating them into the cavity is necessary. This should include direct measurements of T_2 , the coherence time, outside of the cavity environment. Furthermore, the impact of decoherence mechanisms on the emitter performance should be evaluated.
- 2) **Tunability and Preservation of Purcell factor**: The authors should demonstrate that the QDs and the cavity can be tuned post-fabrication while maintaining a F_p above 5. This would show the potential for practical applications where such tunability is essential.
- 3) **Charge Injection Integration**: The incorporation of charge injection mechanisms could potentially increase the utility and function of the device, while providing a reversible tuning mechanism.

In conclusion, the current form of the paper appears incremental, considering the advancements previously published both by the authors and by others in the field. More efficient single-photon sources operating near 1550 nm have been demonstrated, along with quantum emitters in the C-band displaying narrower inhomogeneous broadening and longer coherence times. While the scientific results presented are robust, they represent

more of an evolutionary development rather than a leap in innovation. As such, the current contribution may not meet the threshold for substantial innovation and broad impact that Nature Communications seeks.

Reviewer #3 (Remarks to the Author):

I have read the Author's responses, reports of the other Referees, and the manuscript one more time.

I find that the authors have answered all the requests satisfactory and I can be recommended the manuscript for publication.

Response to Reviewers' comments

We sincerely thank the Reviewers for their time and efforts, and we greatly appreciate their recognition of our work and constructive comments. Below, we give a detailed response to their reports. To enhance clarity, we introduced a coloring scheme outlining **the comments given during the second round in red**. In contrast, the **comments from the first round and our responses from the first round are given in blue**. The sentences added to the manuscript are indicated **in green**.

Reviewer #1 (Remarks to the Author):

I find that the authors' responses to all my comments are satisfying, though I have a couple of additional small comments below that should be addressed. I think all the changes the authors made to the main text and SI are good.

We are glad to hear that we have addressed the comments of Reviewer #1 in a satisfactory manner.

Regarding the response to my Comment 2: The authors claim that non-radiative paths created by structural defects created during growth, or induced by etching. I think the authors should expand on how these factors are not show-stoppers for this integration platform. It sounds a bit like these are things that cannot be avoided - is that the case, or there are reasonable, rational paths forward toward improving defects?

During the first round of the review, Reviewer #1 asked us why the maximum experimental collection efficiency of 16.6 % is considerably lower than expected from simulations. At that time, we responded:

The method for estimating photon extraction efficiency assumes that the QD has an internal quantum efficiency of 100% (...)

And then, we provided the explanation that

[the] discrepancy can be attributed to the non-radiative recombination channels introduced to the QDs due to structural defects propagating from the InP substrate and/or defect states at the side walls of the CBG central mesa, which are introduced during dry etching. These defects most likely cause additional exciton energy

¹ The original manuscript title: *Scalable quantum photonic devices emitting indistinguishable photons in the telecom C-band*

relaxation channels. These are, however, difficult to account for and hence not considered in the model.

After we had finished the characterization of the devices, the quality of the InP wafer used for the epitaxial growth was questioned by the results of another ongoing project: the density of structural defects has been too large and apparently did not comply with the values specified by the producer. Therefore, one has to be particularly careful when choosing the supplier and wafers to grow QDs. In general, we believe that the InP platform is of obtaining the same high material quality as obtained with GaAs, as evidenced by the recently published results on InAs/InP QDs with the conclusion that the coherence time of scattered photons is at least equal to the Fourier limit [34].

The impact of the defects introduced during the dry etching process can be lowered by using a thicker hard mask during the ICP etching of InP. This will prevent ions from penetrating the QD matrix and is feasible with the CBG design, where the critical dimension is ~300 nm. Furthermore, we are actively working on surface state passivation of the cavities to avoid non-radiative relaxation via surface mid-gap states and to reduce the impact of charge noise on the device performance.

Therefore, neither the InP material platform nor the dry etching technique possesses fundamental limitations that would be show-stoppers for our approach.

We included this explanation in the Discussion section:

The discrepancy between the simulated and measured photon extraction efficiency is attributed to the residual defects in the epitaxial material and possible material damage due to the dry etching. The fabrication can be optimized to eliminate both effects.

[34] L. Wells, T. Müller, R. M. Stevenson, J. Skiba-Szymanska, D. A. Ritchie, and A. J. Shields, "Coherent light scattering from a telecom C-band quantum dot," Nat. Commun. 14, 8371 (2023).

I agree with referee #2 that the word "scalability" in the title can be somewhat misleading, or at least there is too much nuance involved that it probably does not belong there. I would suggest maybe replacing the word with "high-throughput", which is essentially what the positioning technique provides in comparison with other, current techniques used for the same purpose. High throughput is necessary but not sufficient for device fabrication scalability, and the term is less nuanced. It's worthwhile pointing out that control of a single emitter spatial location is a tall hurdle towards scalability for any solid state quantum emitter material system, and is yet to be overcome. To some extent, heterogeneous optical properties is a less severe problem for e.g., color centers, however still also a barrier towards device scalability.

We are thankful for raising doubts about whether the term *scalable* is used correctly in the title of our manuscript. After receiving similar feedback from Reviewers #1 and #2, we decided to change the title of the manuscript and replace the term *scalable* with *high-throughput*. We believe that the new adjective summarizes better the content of our paper and the perspectives that our results indicate.

After the authors address these small comments, my opinion is that the manuscript is acceptable for Nature Communications.

We are grateful for the positive recommendation. We would like to once again thank Reviewer #1 for the time he or she has spent on improving our manuscript by providing us with valuable feedback. His/her comments have substantially contributed to enhancing the quality of our manuscript.

Reviewer #2 (Remarks to the Author):

The reviewer's comment during the first round: 1) However, as elaborated below, the results are mainly incremental with low potential to advance the field. As such, publication in Nature Communications is NOT recommended.

Our reply to the comment during the first round: We politely disagree with the Reviewer's judgment. Telecom C-band quantum photonic device deterministic and scalable fabrication with 30% yield and observation of HOM interference are substantial advancements in the field, motivating and inspiring further efforts in quantum photonics research at the telecom wavelength.

The reviewer's comment during the second round: QDs emitting around 1550 nm have been a topic of research for at least the last 20 years. There have already been many reports in literature on the realization of efficient single photon emission with high purity using InAs/InP or InAs/GaAs structures. In InAs/InP QDs, values of $g(2)(0) \sim 4 \times 10^{-4}$ have been reported [e.g., T. Miyazawa et al., Appl. Phys. Lett. 109(13), 132106 (2016)], and secure key transmission over long distances (> 100 km) through optical fiber has been achieved. Single photon sources are characterized by $g(2)(0) \sim 0$ and photon indistinguishability.

Our reply to the comment during the second round: The impressive results obtained by other groups working with QDs emitting around 1550 nm are inspiring. However, our intention and message are not to improve the performance value of previous demonstrations, e.g., higher purity of single-photons, which is already limited by experimental conditions rather than the source itself. All previous results on 1550 nm QD emission with impressive performance values were obtained from randomly assembled QD-photonic structures, i.e., it has been a matter of tedious search in the final characterization and luck to identify a structure with the sought properties. Of course, this "random fabrication" approach does not scale, as stated by the Reviewer in one of his/her comments.

Our work, in strong contrast to all previous results on QD devices at 1550 nm, **for the first time demonstrates the fabrication of QD photonic devices with a yield of 30% operating in a narrow spectral window around the important wavelength of 1550 nm.** Despite succeeding statistically with $\frac{1}{3}$ of all devices, they are characterized by a very high photon purity and indistinguishability, which are standard merits used in the community. The high device throughput is enabled by the **combination of material quality grown by us, sample design, and imaging in the NIR spectral range.** This set is unique and allows us to address the main limitations of previous QD-device fabrication processes, random nucleation, and inhomogeneous broadening (which, according to the Reviewer, have hindered progress).

Recently, a few outstanding articles tackling the main challenges of QD-based single-photon sources at 1550 nm have appeared in high-impact journals, including Nature Communications, proving that despite >20 years of research, it is a captivating subject that sparks significant interest in the community. Among others, impressive articles published in Nat. Comm. started to appear in 2018 by demonstrating the generation of polarization-entangled photon pairs by the XX-X cascade [T. Müller et al., *Nat. Commun.* 9, 862 (2018)], followed in 2022 by an optically active solid-state spin-qubit based on a hole confined in a single InAs/GaAs quantum dot grown on an InGaAs metamorphic buffer layer [Ł. Dusanowski et al., *Nat. Commun.* 13, 748 (2022)], and in 2023 by showing the emission of photons from InAs/InP QDs with coherence times much longer than the Fourier limit via elastic scattering of excitation laser photons [34].

Challenges remain and have to be overcome to advance quantum dot-based technologies. These include, as also mentioned by the Reviewer and, despite >20 years of research, considerable inhomogeneous broadening, low brightness of single-photon sources, lack of inherent scalability due to self-assembly of QDs, and limited possibility for tuning the QD states.

The novelty of our deterministic fabrication, which has been until now been challenging for QDs emitting in the 1550 nm spectral range, opens possibilities to address the enlisted obstacles, thus opening prospects to yield functional devices applicable in quantum photonics. Our method offers quick pre-selection of a QD emitter and shaping of its photonic environment. It provides wavelength selectivity in the pre-selection process and brightness enhancement due to cavity integration, offering the fabrication of many identical devices in a single process run. Therefore, we initially used the term *scalable*, and now, *high-throughput* process.

However, the presented proof-of-concept devices and fabrication methods have more to offer. The QDs can be placed in a *p-i-n* junction and integrated with, e.g., piezoelectric actuators, offering the possibility for spectral tuning and enhancing the device's overall performance, including photon indistinguishability and coherence.

[34] L. Wells, T. Müller, R. M. Stevenson, J. Skiba-Szymanska, D. A. Ritchie, and A. J. Shields, "Coherent light scattering from a telecom C-band quantum dot," *Nat. Commun.* 14, 8371 (2023).

The reviewer's comment during the second round: HOM measurement has been performed using either CW or pulsed excitation [e.g., C. Santori, D. Fattal, and Y. Yamamoto, "Indistinguishable photons from a single-photon device," *Nature* 419, 594 (2002)].

Achieving a 30% yield is commendable, but it is more of an incremental advancement rather than an innovative breakthrough.

Our reply to the comment during the second round: HOM measurement has indeed become a standard characterization technique for QDs (and other quantum light sources) since the breakthrough demonstration cited by the Reviewer. However, the interference contrast in a HOM experiment largely depends on the investigated material. With QDs made of InAs/GaAs operating in the wavelength range of 900-950 nm, close to 100% interference contrast values have been reported. For the InAs/InP QD system operating in the 1550 nm wavelength range, the HOM interference contrast reported by our team in this work is

a record value. Obviously, the HOM contrast value has to be improved for technological applications, and we outline how this can be done. We believe that reporting the progress towards a source with ideal properties is valuable for the community and deserves publication in a high-impact journal. Again, we emphasize that this has been achieved with a device fabricated deterministically.

The first-ever fabrication of telecom C-band QD devices with a yield of 30%, as reported in our work, is not just an incremental advancement but a major achievement. All previously reported results on telecom C-band QDs, including the ones mentioned by the reviewer, have been obtained with random placement and an actual device yield $\ll 1\%$, obviously providing no perspective for repeatability. In our case, we can select the QDs with the desired transition wavelength *before* the photonic device fabrication. As the selection favors brighter QDs (surface density sufficient for technological applications), we also acknowledge a correlation between the deterministic fabrication and the obtained HOM interference contrast.

The reviewer's comment during the first round: 3) The authors gave a certain emphasis (title) to the scalability of their approach. Scalability requires a clear context for better understanding and accurate interpretation. Without such knowledge, determining the scalability of the structure remains ambiguous. Here, it is unclear if the structure needed to optimize the localization (Fig. 1a and Fig. 2b) is tolerant with the implementation of real quantum devices in terms of scalable integration and manufacturing. Scalability in terms of integrating other functional photonic devices would rely on the compatibility of structures and materials. In this regard, the authors should show how that QD-CBG structure in presence of metal markers can be coupled to a heterogeneous nanophotonic structure. Thorough analysis and testing should be conducted to ensure seamless interaction and optimized performance.

Our reply to the comment during the first round: The heterogeneous integration of the QD-CBG structure has never been our aim.

The reviewer's comment during the second round: I am glad the authors acknowledge their structure's integration limitations, which is a critical barrier to its potential. Nevertheless, in their reply, they propose integrating the QD-cavity into a p-i-n diode to enable carrier injection, indicating an understanding of the necessity for integration in enhancing device functionality.

Our reply to the comment during the second round: The application of the *p-i-n* junction in the way suggested by the Reviewer differs from our intentions and explanations provided during the first review round. At that time, we indeed mentioned the *p-i-n* several times, but we wrote:

In the next generation of devices, we envision the implementation of a *p-i-n* junction **to reduce the charge noise** in the vicinity of the QDs and to control its charge state (see also answer to Comment 2, Reviewer #2).

To mitigate the effects of the fluctuating charge environment, we plan to introduce techniques such as further reducing the density of QDs or placing QDs in the p-i-n junction to control the QD charge state in the next generation of our devices.

To mitigate this problem, one can passivate the cavity sidewalls or employ the p-i-n junction to **control the charge environment** in the surrounding QD material.

In our case, **we expect the p-i-n structure to reduce charge noise in the QD vicinity** (Refs. [7] and [35]), thus increasing the photon coherence time and enabling electrical field tuning via the Stark effect.

Additionally, our ongoing efforts to embed the QDs in a **p-i-n junction will provide control over the exciton configuration** so that the QD is depleted of resident carriers and exciton and biexciton dominate the μ PL spectrum if needed.

Specifically, enabling the emitter state preparation through carrier injection is not currently our aim. It can negatively impact device performance concerning triggered and fast operation, photon indistinguishability, and coherence, among others, simply due to generating much more excess charges in the QD vicinity.

Advantages of optical QD pumping

On the contrary, the state preparation through optical pumping can be brought closer to real-life applications using, e.g., hybrid integration, in particular, fiber coupling, following the demonstration of A. Musiał et al., *Adv. Quantum Technol.*, 3, 2000018 (2020). Moreover, the recently introduced SUPER scheme [T. Bracht et al., *PRX Quantum*, 2, 40354 (2021)], provides prospects to lead to unprecedented fidelity of the emitter state preparation, which is beneficial for source coherence. Additionally, the detuned laser pulses in the SUPER scheme allow for straightforward spectral filtering, promising for in-fiber applications.

Optical pumping is very often the preferred option also because it can allow choosing the polarization of the emitted photon (relevant, e.g., in the demultiplexing applications), e.g., by using stimulated two-photon excitation [J. Yan et al., *Nano Lett.* 22, 4, 1483-1490 (2022)]. This also allows switching the emitted photons between pure and mixed states in the photon number basis [Y. Karli et al., *Npj Quantum Inf.* 10, 1 (2024)]. Moreover, there are ways to generate entanglement that require not only the excitation of the QD but also the controlled de-excitation [S. Wein et al., *Nat. Phot.* 16, 5, 374-379 (2022)]. Finally, other schemes are used to create, for example, cluster states using optical pulse sequences [I. Schwartz et al., *Science* 354, 6311 (2016)], or by driving the QD into specific states, which only work with specifically tailored laser pulses [F. Kappe et al., *Adv. Quantum Technol.*, 2300352 (2024)].

In summary, optical excitation provides more freedom as one can create pulse sequences of different pulse areas, combine differently detuned pulses, or apply chirp, leading to coherent control of the system, which is impossible under only electrical driving.

The p-i-n junction

With the “p-i-n junction integration,” we refer to the fabrication of a monolithic structure, where a QD layer is sandwiched between highly p- and n-doped InP regions, respectively.

Such a structure provides a constant electric field distribution across the QD layer, accumulating free carriers at interfaces far from the QD, diminishing charge fluctuations, and allowing control of the QD charge state. These can be applied to reducing the emitter linewidth, otherwise broadened by fluctuations in the charge environment, and to tuning the QD emission wavelength utilizing the Stark effect. Fundamentally, reducing the linewidth will increase coherence and thus improve indistinguishability. We emphasize that the demonstrated deterministic fabrication scheme also applies to any QD structure in a *p-i-n* junction. Besides the location, QDs can then also be validated according to optical and electronic requirements.

Therefore, we argue that we have diverging opinions on *the necessity for integration in enhancing device functionality*, as according to the Reviewer, the electrical carrier injection is necessary for improving the device functionality. We suppose that the different views on how to develop the platform further are one of the reasons for the Reviewer's concern regarding the word *scalability* (which we have replaced with the adjective *high-throughput_now*).

Nevertheless, we believe that this dispute concerning purely future development should not be reflected in the overall negative opinion of our current work.

Our reply to the comment during the first round: Our understanding of the term *scalability* refers to the QD localization process.

The reviewer's comment during the second round: Localization of QDs and their coupling with cavities was reported almost 20 years ago, but the QD-cavity system has not been considered 'scalable' by the community working in this field.

This is because the QDs nucleate randomly on the surface and the large inhomogeneous broadening inherent in the self-assembly process. This means that:

- i) each QD must be located separately,
- ii) each cavity must be designed with a different resonance.

Only a structure with spatially ordered QDs and small inhomogeneous broadening can be considered a scalable system in the sense meant by the authors. For instance, site-controlled InGaAs pyramidal QDs exhibit such characteristics, even though they come with other issues.

Our reply to the comment during the second round: We respect the understanding of the term *scalable* by Reviewer #2. We have removed it from the title to address the concerns regarding the implications of using this term.

Once more, we would like to direct the Reviewer's attention to the QD localization technique presented in our manuscript, which offers pre-selection of the emitters just at the PL imaging step. Effectively, this technique significantly mitigates the QD ensemble inhomogeneity. At the end of the PL imaging step, only the subset of QD emitters is selected with narrow specifications with respect to the emission wavelength, the emission line broadening, and the intensity. The remaining amount of inhomogeneity only depends on the linewidth of the

filter. Therefore, **QDs with close to identical properties are selected in large quantities in parallel using our approach**, not one by one as claimed by the reviewer. Consequently, the cavity geometry does not have to be adjusted for each QD if the cavity resonance is broader than the filter linewidth and can be fixed according to the requirements (as done in our work).

Even for two-photon interference from different/remote QDs, we disagree with the Reviewer's point ii) that *each cavity must be designed with a different resonance*. For two or more QDs, one would use the same cavity design (or photonic nanostructure in general) for every QD but then fine-tune each QD with respect to the cavity resonance. Only if the cavity resonances are also inhomogeneously broadened would tuning the cavity resonances be required. However, this is far beyond the scope of the present work and discussion.

In the first round of the review process, we extensively explained our point of view and provided a viable route for possible industry-like device fabrication (pp. 20-21). Still, we would like to take this opportunity and advocate again for our differing opinions. First, the Reviewer claims that *the QD-cavity system has not been considered 'scalable' by the community working in this field*. We believe that emphasizing the scalability prospects was simply not the priority of other groups, except for some notable exceptions, for instance, the realization in the group of Pascale Senellart and the start-up Quandela. The company has commercialized cavity-coupled Stranski-Krastanov QDs, which are first localized in **PL imaging**. We wrote in the first round of the review:

Our reply to the comment during the first round: A similar concept with a localization approach was applied in Ref. [28] and has become the foundation of the Quandela company which, i.a., sells fiber-coupled single-photon sources operating at ~ 900 nm.

Our understanding is that if one can successfully run a start-up based on the localization of QDs that have a considerable inhomogeneous broadening, the approach must be **sufficiently scalable**. Therefore, the concerns raised by Reviewer 2, specifically that *i) each QD must be located separately, ii) each cavity must be designed with a different resonance* can be addressed.

Our reply to the comment during the first round: What has hindered the progress in the field so far is the scalability of embedding a QD in a photonic device, not the scalability of integrating the fabricated nanophotonic device with a QD with the target platform.

The reviewer's comment during the second round: Again, the localization of QDs and their coupling with cavities was reported almost 20 years ago. What has prevented the field from moving forward is the large inhomogeneous broadening inherent in self-assembled QDs, that is, QDs emit single photons at different wavelengths.

For the specific device the author has in mind, a standalone single-photon source, a key requirement is low uncertainty in the emission wavelength. For example, to scale photonic systems that require parallel photon generation, these sources must emit at uniform

wavelengths, or the inhomogeneous broadening should be significantly less than the intrinsic linewidth.

Our reply to the comment during the second round: Again, in a parallel approach, we select QDs within a narrow spectral range around 1550 nm. Although the overall distribution of QDs remains inhomogeneously broadened, the distribution of QDs we identify in the subset with ~8 nm filter bandwidth is significantly narrower. This is within the CBG linewidth of ~8 nm in the present case.

Indeed, 20 years ago, this selection process was not feasible, and QD-cavity pairs had to be selected one by one, requiring major efforts in practice using, e.g., an AFM. Our growth process yields a QD density of $\sim 4 \times 10^5 \text{ cm}^{-2}$ with a wavelength within 8 nm bandwidth. Even filtering with <1 nm bandwidth would leave a significant number of QDs for parallel fabrication of single-photon devices with close to identical properties (that we intend to fine-tune in future projects using *p-i-n* junction; see other comments). We are confident that with this, our approach supports significant prospects for realizing stand-alone parallel photon sources emitting at a uniform wavelength in the 1550 nm range. Therefore, the inhomogeneous broadening is no longer an obstacle, as the Reviewer incorrectly claimed.

Of course, reducing the overall inhomogeneous distribution will be an additional benefit. Still, it remains challenging to achieve this with self-assembled QDs emitting around 1550 nm (see, for example [R. Sittig *et al.*, *Nanophotonics* 11, 6, 1109-1116 (2022)], where the broadening is comparable to our InAs/InP system). An alternative approach would be deterministic and site-selective growth, but the properties of such QDs are unsatisfactory thus far, leaving self-assembled QDs with no alternative at the moment.

The reviewer's comment during the first round: 6 - Relatedly, have the authors attempted LA-photon or direct resonant excitation of the QD? It would be interesting to see if the coherence time can be significantly improved, since these two excitation methods are minimally detrimental to the single-photon coherence times.

Our reply to the comment during the first round: We attempted LA-phonon and strict resonant excitation, which would indeed give further insights into limiting factors. However, the necessary polarization filtering of the reflected laser light under resonant excitation is more challenging for micro- or nanophotonic structures such as CBGs than for planar samples due to the scattering of laser light. We did observe spectral signatures of resonance fluorescence. Unfortunately, HOM experiments have not been within reach under this excitation scheme due to the remaining amount of unsuppressed scattered laser light.

The reviewer's comment during the second round: Resonant scattering in single QDs and in single QDs coupled to microcavities has been documented in many papers now. It has been reported even in structures (such as 2D L3 photonic crystals) that present a mode-mismatch with the incoming gaussian laser beam much larger than in CBGs.

Our reply to the comment during the second round: Indeed, the resonant fluorescence of QDs has become more widespread in recent years. This circumstance, however, does not

change the fact that strictly resonant or quasi-resonant excitation of quantum emitters will always remain a troublesome and very sophisticated experiment due to the need to suppress the scattered laser light with a high extinction ratio, $\gg 30\text{dB}$. This is a well-developed technique for quantum light sources with a wavelength $<1\ \mu\text{m}$ (which we believe the reviewer refers to), but for technical reasons, as of now, this remains more challenging around $1.55\ \mu\text{m}$ wavelength. Therefore, for now, we are satisfied to have observed the spectral features of resonance fluorescence during the investigation of the current devices, and we regret that we have not achieved the laser suppression sufficient to perform the HOM experiments under these excitation conditions. Such experiments, however, have never been a principal purpose of the undertaken research.

The reviewer's comment during the first round: The demonstrated positioning involves CBG, which possesses a quite large mode volume (V) and a low-quality factor (Q) (fig 3a). While this is useful for showcasing deterministic coupling, the most crucial quantum devices necessitate an ultra-high Q/V ratio. This is evident also in the low Purcell factor reported.

Our reply to the comment during the first round: There are several crucial parameters for a cavity-coupled single-photon source, and some trade-offs between them are necessary to fulfill the device specification for a particular purpose.

The reviewer's comment during the second round: The CBG has a large mode volume, and the reported Purcell factor (F_p) is clearly quite low. F_p is proportional to Q/V . Because the authors focus on devices based on the weak coupling regime (such as single photon sources), they should still show a sizable Q/V . High efficiency is very important, but several emitter-cavity structures have already been shown to maintain high Q/V while achieving high vertical extraction (as seen in micropillar configurations) or lateral extraction (as integrated with 2D photonic crystals).

Our reply to the comment during the second round: The fabricated CBGs have a smaller mode volume than micropillars. Also, we are unsure what the Reviewer implies with a *sizable Q/V* . The Q/V ratio is not benchmarked across the literature for single photon sources that rely on the QD-cavity architecture. For the presented architecture with a QD in a CBG cavity with emission in the $1550\ \text{nm}$ spectral range, our results demonstrate the record Purcell factor provided by the Q/V ratio specified for our system. The discussion on the Q/V ratio and, thus, the Purcell factor cannot be complete without the context of the material system, respective accessible technological processes, and the cavity design, as well as the functionality of the final device. **The results presented here, the technological processes applied to the InAs/InP material system with the InAs QD in a CBG cavity emitting at $1550\ \text{nm}$ are at present state-of-the-art, and the achievable device parameters are the best in their class.**

Our reply to the comment during the first round: It is known that high- Q and low- V cavities can achieve the strong coupling condition. Once reached, the source can operate in a photon blockade mode, effectively suppressing multi-photon events,

thus providing photon generation with high purity [K. Müller et al., Phys. Rev. Lett. 114, 233601 (2001)]. In addition, the strong coupling condition can considerably speed up the emission, switching off most phonon-mediated decoherence channels, leading to high photon indistinguishability—the source's high speed and coherence suit quantum communication and quantum computation applications. However, a strong interaction of an emitter with the cavity mode opens new decoherence mechanisms related to the excitation of higher states in the Jaynes-Cummings ladder, deteriorating the source coherence. Again, there is a trade-off between the coupling regime and the required coherence level of the source. Because of that, implementing the weak coupling regime can be sufficient.

The reviewer's comment during the second round: This paper aims to demonstrate efficient single-photon emission. I surely agree that it is not beneficial for an emitter-cavity system meant for single-photon generation to be far into the strong coupling regime, and I did not mean that. Increasing Q/V will not necessarily bring one into the strong coupling regime.

Our reply to the comment during the second round: Also here, we kindly disagree with the Reviewer. **Our article is not primarily focused on efficient single photon emission.** The context is much broader and tackles deterministic and high-throughput fabrication of devices that rely on an InAs/InP QD emitting at 1550 nm coupled to the InP CBG cavity, demonstrating state-of-the-art performances for such devices. Beyond that, we also demonstrate that the emitted photons are characterized by indistinguishability, which is merit for the high quality.

Our reply to the comment during the first round: Consequently, high Q factor cavities reduce the source brightness.

The reviewer's comment during the second round: To a certain extent, because the brightness will be also improved by the enhanced spontaneous emission, which means high F_p .

However, a crucial figure of merit is the overall out-coupling efficiency, which measures how effectively internally generated single photons are transmitted into a desired single mode. This efficiency approaches one when F_p is large and there is minimal loss of cavity photons to modes other than the desired one.

Our reply to the comment during the second round: We agree that the outcoupling efficiency is very important, and according to [L. Bremer *et al.*, *Opt. Express* 30, 10 (2022)], the CBG cavity can provide high overall outcoupling efficiency, even higher than the micropillar cavity under relaxed coupling conditions for the emitter-cavity field.

Our reply to the comment during the first round: Contrary to the Reviewer's opinion, we believe that a CBG is an interesting photonic structure that features high levels of photon extraction efficiency in a broad spectral range (Fig. 1b and Ref. [14]). Moreover, the CBG is effectively an antenna structure with moderate Q factors (110

in our case) that results in Purcell enhancement with a spectrally broad response (Ref. [14]; here, FWHM of 14 nm).

The reviewer's comment during the second round: CBG is an interesting structure that has been studied for many years. However, I do not believe that the findings presented in this paper, considering what is already in the literature (including publications by the authors), are substantial enough to warrant publication in Nature Communications.

Our reply to the comment during the second round: We respect the Reviewer's opinion on the degree of novelty of our results and regret that we were unable to change his/her mind during the review process. **Thus, we can only stress once again that the first-time development of deterministic fabrication technology for cavity-enhanced QD devices operating at telecom C-band wavelengths is an important milestone, opening the route for substantial advances in the field and toward applications in practical quantum information science. The CBG is the object we chose to demonstrate the power of our fabrication technique, and we do not claim any novelty on this part.** Hence, there can be no doubt from our point of view that our work is well suited for the Nature Communications journal and highly interesting for the entire readership. To clarify these points further, within the second round of revisions, we added an extended discussion to our manuscript detailing future prospects for integration and tunability.

The reviewer's comment during the first round: This work primarily builds upon the existing literature and previous publications from the same group. (i) The positioning of QDs to photonic resonators (such as CBG) has been demonstrated multiple times using the same PL technique. (ii) Besides the SPS (already showed by several groups and in ref. 10 by the authors), this paper has not demonstrated the versatility of this structure in enabling new quantum devices. For example, it is unclear if the CBG can be tuned independently from the QDs or vice versa. Additionally, questions remain regarding the possible implementation of quantum optics schemes ... or the ability to voltage-control the QD exciton or trigger emission. In this regard it is not a breakthrough that significantly extends the knowledge and fosters innovation, ultimately contributing to the development of new technologies, quantum protocols, or device research.

The reviewer's comment during the second round: The authors' reply did not adequately address my criticisms above. It is crucial to achieve fine, independent tuning, whereas Figure R6 indicates that the device exhibits only an irreversible, coarse adjustment. Several research groups have demonstrated reversible, independent tuning of the emitter and cavity using techniques such as the Stark effect (for the emitter) and layer deposition (for adjusting the cavity).

Our reply to the comment during the second round: Layer deposition is a method that irreversibly tunes the cavity resonance [T. Krieger, *Postfabrication Tuning of Circular Bragg Resonators for Enhanced Emitter-Cavity Coupling*, *ACS Photonics* (2024), accepted]. Regarding reversible independent tuning of the emitter, we have discussed this extensively (p-i-n junction/Stark effect or strain tuning) and provided our response at the conclusion of the Reviewer's 2 remarks.

The same authors [ac]knowledge the necessity to implement an electrical tuning combined with carrier injection:

Our reply to the comment during the first round: “*In the next generation of devices, we aim to implement electrical control of the QD emission. This requires a p-i-n structure such as, for example, applied in the InAs/InP QD system to electrically inject the carriers.*”

Our reply to the comment during the second round: As explained above, the *p-i-n* junction, in our opinion, should not serve for charge injection as the means of emitting state preparation. After the second cited sentence, we provided a reference to [T. Müller et al., *Nat. Commun.* 9, 862 (2018), Z.-H. Xiang, *Commun. Phys.* 3, 121 (2020)]. In the first cited article, the carriers were indeed electrically injected into the QDs. However, the authors do not report on the photon indistinguishability, which is expected to be very poor in this configuration. The citations served as examples of the *p-i-n* junction formed in the InP system, and we regret that it suggested that we intend to use a *p-i-n* junction to inject the carriers in the QDs, in a similar manner to [T. Müller et al., *Nat. Commun.* 9, 862 (2018)].

The reviewer's comment during the second round: I encourage the authors to further develop their proposed device to clearly demonstrate its versatility and potential advantages.

Our reply to the comment during the second round: The platform's further development will surely follow the current demonstration. However, the main message of the present manuscript is the deterministic fabrication of photonic devices of nearly any desired kind, sufficiently well proven in the current form of the manuscript.

The reviewer's comment during the second round: However, I have concerns regarding the integration of a *p-i-n* structure within the current design. I suspect that this addition may further reduce the quality factor (Q) of the device due to the absorption by free charges.

Our reply to the comment during the second round: We are not planning to use the *p-i-n* junction as a charge injection method but to stabilize the electric field across the QD layer, as we have already explained. Reference [Q. Buchinger et al., *Appl. Phys. Lett.*, 122, 111110 (2023)] reports a numerical optimization of a **CBG with contact bridges and experimentally verifies that with contacts, Q-factors > 200 are feasible (compared to Q ~ 194 of our fabricated cavity)**. Again, we have not claimed to restrict our future efforts to CBGs.

The reviewer's comment during the second round: To critically enhance the significance of this paper and align it with the standards expected by Nature Communications, the authors should consider the following recommendations:

1) ****Comprehensive Characterization of Emitters****: A detailed study of the emitters before integrating them into the cavity is necessary. This should include direct measurements of T₂, the coherence time, outside of the cavity environment. Furthermore, the impact of decoherence mechanisms on the emitter performance should be evaluated.

Our reply to the comment during the second round: We have constructed an all-fiber-based Michelson interferometer (MI) within the second round of revisions. Response Figure R1 depicts the dedicated Michelson setup as well as exemplary measurement data obtained from CBG device #2, from which we extract the T_2 time. Most importantly, the additional experimental data obtained via MI measurements reveal no noticeable degradation of CBG-integrated QDs compared to emitters in unprocessed sample regions. The obtained coherence times T_2 are comparable to those reported in the literature for similar QDs.

Figure R1: (a) Schematic of the all-fiber Michelson interferometer for 1550 nm. The single photon signal is split in a fiber beamsplitter (BS) and back-reflected at two Faraday mirrors (FM), while the coarse delay on one arm is controlled with a variable optical delay line (VDL), and the fine scan of the relative path difference is done via a piezo fiber stretcher (FS). The counts after the interference are detected on a superconducting nanowire single photon detector (SNSPD). Inset: Example of the FS scan for 0 ps delay indicating constructive and destructive interference. (b) Normalized fringe visibility for the QD-CBG #2 under 980 nm CW excitation. Extracted visibility for different delay positions allows the determination of the T_2 time using an exponential fit.

The corresponding experimental results, supporting our previous findings, are summarized in a new paragraph within the section “Photon coherence time” of the manuscript and the experimental details in the Methods section. More details on the experimental results are added to the new section E of Supplementary Note 7.

Text added within the section “Photon coherence time”:

To gain further insights into the coherence properties, we performed direct measurements of the T_2 time using an all-fiber-based Michelson interferometer (MI; see Methods). We extract a coherence time of up to (62 ± 3) ps for the QD-CBG #2 under weak CW above-band excitation (see Supplementary Note 7). Note that while this value is lower than the T_2 time extracted from the dip in the HOM experiment in Fig. 4c, a direct comparison is not possible due to the different excitation schemes applied. An analysis of MI data from a total of three different CBG devices yields T_2 values between 18 ps and 60 ps. These numbers compare favorably with MI-measured coherence times of 6-30 ps obtained for three different QDs in

planar regions on the same sample. The observed coherence times are comparable with reports in the literature for SK InAs/InP QDs²². While the direct comparison of the T_2 values measured for QDs with and without CBG should be treated with care due to the relatively low statistics, these results indicate that the microcavity integration does not degrade the optical coherence of the emitted photons. Future work in this direction may include a more elaborate study allowing for deeper insights into the limiting dephasing mechanisms and their timescales, e.g., by applying photon-correlation fourier-spectroscopy²⁴.

Text added at the end of the Methods section:

Coherence measurements - For measurements of T_2 time, an all-fiber-based Michelson interferometer (MI) was implemented²², consisting of a 2x2-port 50:50 fiber beam splitter with both exit ports terminated by a Faraday mirror, reflecting the light with 90° polarization rotation. The necessary coarse and fine temporal delay is controlled by a variable optical delay stage and a piezo-driven fiber stretcher in the two MI arms, respectively. The single-photon signal is coupled to one input port of the MI and detected at the second input port using a SNSPD. The MI setup in its configuration features 80% overall transmission (excluding the BS) and allows for the measurement of coherence times of up to 1 ns. The maximally achievable interference contrast was measured with a CW laser at 1550 nm to be 98%, limited only by the slight intensity mismatch due to the reduced transmission through the optical delay line. For each temporal delay adjusted via the coarse variable delay line, a fine temporal scan is performed via the fiber stretcher, resulting in interference fringes with an amplitude depending on the overall delay. The interference fringes are evaluated by subtracting a constant amount of dark counts and evaluating the interference contrast via $v = (I_{max} - I_{min}) / (I_{max} + I_{min})$. Finally, the T_2 time is extracted by fitting a two-sided exponential decay to the interference visibility v data as a function of the coarse delay set in the MI with the uncertainty representing the fit accuracy.

Text added into the Discussion section:

The photon coherence time measured using a Michelson interferometer is up to (62 ± 3) ps for the QD-CBG #2 under weak CW above-band excitation.

Additionally, Supplementary Note 7 was extensively expanded. We added section E (pp. 19-22).

[22] M. Anderson, T. Müller, J. Skiba-Szymanska, A. B. Krysa, J. Huwer, R. M. Stevenson, J. Heffernan, D. A. Ritchie, and A. J. Shields, "Coherence in single photon emission from droplet epitaxy and Stranski–Krastanov quantum dots in the telecom C-band," Appl. Phys. Lett. 118, 014003 (2021).

[24] X. Brokmann, M. Bawendi, L. Coolen, and J.-P. Hermier, "Photon-correlation Fourier spectroscopy," Opt. Express 14, 6333–6341 (2006).

The reviewer's comment during the second round: 2) **Tunability and Preservation of Purcell factor: The authors should demonstrate that the QDs and the cavity can be tuned post-fabrication while maintaining a F_p above 5. This would show the potential for practical applications where such tunability is essential.**

Our reply to the comment during the second round: There are two major advantages of tuning the QDs:

1. lowering the fine structure splitting of the exciton and
2. shifting the QD emission wavelength, mostly to match the cavity mode energy.

The reports on tuning the QDs concern predominantly the first point, while we believe the Reviewer would like to see the wavelength tuning in our devices as an alternative to overcoming the inhomogeneous broadening of QDs, as this issue was raised a few times by her/him.

The reports on the QD energy tuning employ mostly the strain [37-38] and quantum-confined Stark effect [39-40]. However, in each case, the tuning range is very limited, below 1 nm [37-39] or up to 2 nm [40]. For instance, in the state-of-the-art demonstration of the quantum interference of identical photons from remote GaAs QDs [39], the ratio of the Stark tuning range (0.4 nm) to the full-width at half-maximum of the emission ensemble (11 nm) is only 3.6 %. This indicates that the approach cannot overcome the inhomogeneous broadening of the QD ensemble peak, even if it is extremely narrow (11 nm). Therefore, QDs of appropriate emission energy must be identified first, and then the emission wavelength can be tuned. The tunability of QD emission wavelength in a range broad enough to cover the inhomogeneous distribution to avoid pre-selection has not been demonstrated so far. Still, it is also not required as elaborated in detail before. However, this is not a show-stopper for the InAs/InP platform. The QD densities are sufficient for efficient device fabrication, and one can simply use a narrower bandpass filter (~2 nm) in the PL localization to limit the range to where the QD emission can be fine-tuned to match each other perfectly. Note that this does not require reducing the inhomogeneous broadening of the entire QD ensemble.

Another motivation for tuning is lowering the FSS, which can also be important for future devices, and demonstrating this tuning will be another milestone. However, for many applications, a quantum light source can work perfectly with a non-zero FSS, for instance, utilizing the CX transition.

Neither of the tuning capabilities (Stark or strain) has been included in the device design since tuneability is not within the scope of this manuscript. Hence, it cannot be demonstrated now, as requested by the Reviewer. Nonetheless, we believe that proving the main points of the manuscript does not necessitate the demonstration of tuning, the inclusion of which being rather a next step than a mandatory feature of the current device.

We included this description by adding a sentence to the Discussion section:

On the other hand, tuning the QD emission energy using strain^{37,38}, or quantum-confined Stark effect^{39,40}, would address the challenge of QD ensemble inhomogeneous broadening by fine-tuning the QD energy to match the cavity mode. The QD tuning is feasible using the reported approach but requires a different cavity design^{41,42}.

[37] K. D. Jöns, R. Hafenbrak, R. Singh, F. Ding, J. D. Plumhof, A. Rastelli, O. G. Schmidt, G. Bester, and P. Michler, "Dependence of the Redshifted and Blueshifted Photoluminescence Spectra of Single In_xGa_{1-x}As/GaAs Quantum Dots on the Applied Uniaxial Stress," Phys. Rev. Lett. 107, 217402 (2011).

[38] K. D. Zeuner, M. Paul, T. Lettner, C. R. Hedlund, L. Schweickert, S. Steinhauer, L. Yang, J. Zichi, M. Hammar, K. D. Jöns, and V. Zwiller, "A stable wavelength-tunable triggered source of single photons and cascaded photon pairs at the telecom C-band," *Appl. Phys. Lett.* 112, 173102 (2018).

[39] L. Zhai, M. C. Löbl, G. N. Nguyen, J. Ritzmann, A. Javadi, C. Spinnler, A. D. Wieck, A. Ludwig, and R. J. Warburton, "Low-noise GaAs quantum dots for quantum photonics," *Nat. Commun.* 11, 4745 (2020).

[40] L. Zhai, G. N. Nguyen, C. Spinnler, J. Ritzmann, M. C. Löbl, A. D. Wieck, A. Ludwig, A. Javadi, and R. J. Warburton, "Quantum interference of identical photons from remote GaAs quantum dots," *Nat. Nanotechnol.* 17, 829–833 (2022).

The reviewer's comment during the second round: 3) ****Charge Injection Integration****: The incorporation of charge injection mechanisms could potentially increase the utility and function of the device, while providing a reversible tuning mechanism.

Our reply to the comment during the second round: We regret that this comment cannot be addressed in the review process of the current manuscript. First, such a possibility should have been well-thought-out during cavity design and included at the beginning of the fabrication, i.e., the epitaxial growth. Besides this, as already mentioned, we believe that the electrical charge injection would drastically reduce the quality of the optical properties of the device, so including it is contrary to our wish but also contrary to the research direction pursued by other groups for years.

We discussed the *reversible tuning mechanism* in the previous point. In general, it is important that the applied voltage does not increase the amount of carriers available for trapping by the QDs.

We expressed our understanding by expanding the Discussion section:

It is thus important to stabilize the QD environment by removing the excess charge carriers from the vicinity of QDs, e.g., by integrating them into a p-i-n junction, which is expected to increase the photon coherence and indistinguishability substantially. [...] Implementing coherent optical pumping schemes, such as two-photon resonant excitation⁴³, also for scalably fabricated devices, while avoiding the excess charge carriers that could originate, e.g., from the electrical QD excitation, is a crucial next step to further improve the photon coherence time and hence indistinguishability⁴⁴.

[43] D. A. Vajner, P. Holewa, E. Zięba-Ostój, M. Wasiluk, M. von Helversen, A. Sakanas, A. Huck, K. Yvind, N. Gregersen, A. Musiał, M. Syperek, E. Semenova, and T. Heindel, "On-demand generation of indistinguishable photons in the telecom c-band using quantum dot devices," *ACS Photonics* (2024).

[44] A. Reigue, R. Hosten, and V. Voliotis, "Resonance fluorescence of a single semiconductor quantum dot: the impact of a fluctuating electrostatic environment," *Semicond. Sci. Technol.* 34, 113001 (2019).

The reviewer's comment during the second round: In conclusion, the current form of the paper appears incremental, considering the advancements previously published both by the authors and by others in the field. More efficient single-photon sources operating near 1550 nm have been demonstrated, along with quantum emitters in the C-band displaying narrower inhomogeneous broadening and longer coherence times. While the scientific results presented are robust, they represent more of an evolutionary development rather than a leap in innovation. As such, the current contribution may not meet the threshold for substantial innovation and broad impact that Nature Communications seeks.

Our reply to the comment during the second round: We would like to take the opportunity to thank Reviewer 2 for his/her involvement in the review of our paper, sharing his/her criticism, and suggesting corrections to the manuscript in both review rounds. We respect the Reviewer's pessimistic opinion on the soundness of our results and greatly regret that we were unable to change his/her perspective with our response in the first round of the review process. We, however, hope that the newly extended discussion in the manuscript highlighting the realistic prospects that our results enable and the extensive response to the reviewers' comments are convincing to justify publication in Nature Communications.

Reviewer #3 (Remarks to the Author):

I have read the Author's responses, reports of the other Referees, and the manuscript one more time.

I find that the authors have answered all the requests satisfactory and I can be recommended the manuscript for publication.

We are thankful for the work of Reviewer #3, for the time he or she has spent on improving our manuscript by providing us with valuable feedback, and for the positive recommendation regarding the publication of our results.

REVIEWERS' COMMENTS

Reviewer #1 (Remarks to the Author):

The authors also mention that the InP substrate platform has the potential to achieve comparable quality to the GaAs platform used for short wavelengths, citing ref. [34], to bolster the general claim of InP as promising substrate for epitaxial growth. I also find this reasonable, considering that the integration effort would be the same for the droplet epitaxy dots that were used in that publication.

In this new version of the manuscript, the authors have added Fourier transform spectroscopy measurements of the coherence time T_2 for a small number of dots in and outside fabricated cavities. The coherence times are quite short (< 60 ps) in either case in comparison with the natural QD lifetimes of \sim ns. Comparably short coherence times are quite often observed in GaAs / InAs SK QDs as well, depending somewhat on growth quality, and tends to be worse for non-resonant excitation. So I don't see short T_2 s being show-stoppers, in the sense that it's not unreasonable to expect improvement in growth leading to comparable coherence times as observed in the GaAs / InAs SK QD system. The results of ref.[34] suggest as much. The comparison of coherence times for dots in unprocessed and processed portions of the sample suggests that no severe degradation occurs that would be visible under non-resonant excitation. This is a somewhat low bar, as resonant excitation might have revealed degradation more evidently, considering it would have featured lesser charge noise contributions. However, I find that the comparison is acceptable at this stage, considering all the integration work that has been demonstrated, and the discussion around encountered material properties. I recommend publication as is in Nature Communications.